# CD36 maintains the gastric mucosa and associates with gastric disease

Miriam Jacome-Sosa [1✉], Zhi-Feng Miao[2,3], Vivek S. Peche[1], Edward F. Morris[1], Ramkumar Narendran[1], Kathryn M. Pietka[1], Dmitri Samovski[1], Hei-Yong G. Lo [3], Terri Pietka[1], Andrea Varro[4], Latisha Love-Gregory[5], James R. Goldenring [6], Ondrej Kuda [7], Eric R. Gamazon [8,9], Jason C. Mills[10✉] & Nada A. Abumrad [1,11✉]

The gastric epithelium is often exposed to injurious elements and failure of appropriate healing predisposes to ulcers, hemorrhage, and ultimately cancer. We examined the gastric function of CD36, a protein linked to disease and homeostasis. We used the tamoxifen model of gastric injury in mice null for *Cd36* (*Cd36*−/−), with *Cd36* deletion in parietal cells (PC-*Cd36*−/−) or in endothelial cells (EC-*Cd36*−/−). CD36 expresses on corpus ECs, on PC basolateral membranes, and in gastrin and ghrelin cells. Stomachs of *Cd36*−/− mice have altered gland organization and secretion, more fibronectin, and inflammation. Tissue respiration and mitochondrial efficiency are reduced. Phospholipids increased and triglycerides decreased. Mucosal repair after injury is impaired in *Cd36*−/− and EC-*Cd36*−/−, not in PC-*Cd36*−/− mice, and is due to defect of progenitor differentiation to PCs, not of progenitor proliferation or mature PC dysfunction. Relevance to humans is explored in the Vanderbilt BioVu using PrediXcan that links genetically-determined gene expression to clinical phenotypes, which associates low *CD36* mRNA with gastritis, gastric ulcer, and gastro-intestinal hemorrhage. A CD36 variant predicted to disrupt an enhancer site associates (p < 10−17) to death from gastro-intestinal hemorrhage in the UK Biobank. The findings support role of CD36 in gastric tissue repair, and its deletion associated with chronic diseases that can predispose to malignancy.

[1] Center for Human Nutrition, Department of Medicine, Washington University School of Medicine, St. Louis, MO, USA. [2] Department of Surgical Oncology, Laboratory of Precision Diagnosis and Treatment of Gastrointestinal Tumors, First Hospital of China Medical University, Shenyang, China. [3] Division of Gastroenterology, Department of Medicine, Washington University School of Medicine, St. Louis, MO, USA. [4] Institute of Translational Medicine, University of Liverpool, Liverpool, UK. [5] Department of Pathology & Immunology, Washington University School of Medicine, St Louis, MO, USA. [6] Departments of Surgery and Cell and Developmental Biology, Vanderbilt University Medical Center and VA Medical Center, Nashville, TN, USA. [7] Institute of Physiology, Czech Academy of Sciences, Videnska 1083, 14220 Prague 4, Czech Republic. [8] Division of Genetic Medicine, Vanderbilt University Medical Center, Nashville, TN, USA. [9] MRC Epidemiology Unit, University of Cambridge, Cambridge, UK. [10] Gastroenterology & Hepatology Section, Departments of Medicine and of Molecular and Cellular Biology, Baylor College of Medicine, Houston, TX, USA. [11] Department of Cell Biology and Physiology, Washington University School of Medicine, St. Louis, MO, USA. ✉email: mjacome@wustl.edu; Jason.Mills@bcm.edu; nabumrad@wustl.edu

In addition to assisting the gut in nutrient absorption through the secretion of acid and digestive enzymes, the stomach influences metabolic homeostasis through releasing hormones, such as ghrelin and leptin that have systemic effects. Gastric functions are regulated by food intake, the vagus nerve, and in an autocrine manner[1]. Gastric ghrelin, released during fasting, acts in the brain to increase food intake and in adipose tissue to promote energy storage[2,3] in addition to having protective cardiovascular actions[4]. Gastric leptin, unlike adipose leptin which is constitutive, responds to feeding and vagal input to regulate short-term energy metabolism[5,6]. Leptin also has positive effects on epithelial cell proliferation[7] and small intestinal growth[8,9]. Other secretions by the stomach include gastrin and somatostatin, with opposite actions on gastric acid secretion and with numerous effects on organs beyond the stomach[10], and gastric intrinsic factor (GIF) essential for intestinal vitamin B12 absorption[11].

As a port of entry for food, drugs, and pathogens, the gastric mucosa is continuously subjected to injury from chemicals and microbes. The organ's ability to recover from insults depends on the renewal of its epithelial cells. Superficial injury of the gastric mucosa, such as occurs after ethanol, can be repaired rapidly while deeper mucosal injury consequent to aspirin or non-steroidal agents (NSAIDS) or *H. pylori* infection heals more slowly[12,13]. Appropriate repair is required to prevent gastritis, ulcers, hemorrhage, and the development of metaplasia[14]. Although cellular metabolism influences cell fate and function and is dysregulated in disease[15–17], its role in maintaining the gastric epithelium is unclear. In particular, fatty acid (FA) uptake and FA oxidation have been shown important for tissue maintenance and repair, most notably by keeping a competent niche of stem cells. Although FA oxidation does not acutely fuel proliferation of stem cells, it is critical for maintaining the stem cells' long-term ability to differentiate[18,19].

The scavenger receptor CD36 was reported to be abundant in stomach tissue[20] and its mRNA identified a decade ago in a microarray of magnetic bead sorted parietal cells[21]. CD36 facilitates FA uptake and FA oxidation so we investigated if gastric CD36-mediated FA metabolism is important for the function and repair of the gastric mucosa.

In addition to facilitating FA uptake, CD36 mediates signal transduction important in metabolic regulation[22] and tissue homeostasis[23,24]. In the small intestine, CD36 is important for optimal secretion of chylomicrons[25], the release of regulatory gut peptides[26] and the protection of barrier integrity[24]. In the present study, we document the influence of CD36 on stomach biology, function, and recovery from injury. The impact of global versus cell-targeted *Cd36* deletion pinpoints to the critical role of tissue lipid metabolism in epithelial renewal. We also identify genetic associations in two large patient databases between the CD36 expression and gastric disease.

## Results

### *Cd36* deletion alters gastric gland morphology, gastric secretions, and increases matrix accumulation.

The CD36 protein level is robust in the mouse stomach, (Supplementary Fig. 1a). Among the forestomach, corpus, and antrum, CD36 is most abundant in the corpus (Supplementary Fig. 1b). Unlike in the small intestine[27], the CD36 level is unchanged by fasting and refeeding (Supplementary Fig. 1c).

In the corpus of wildtype (WT) mice, gastric glands line up alongside vessels that run from the lumen to the base of the epithelium. In *Cd36*−/− mice, there is frequent misalignment with inconsistency in the relative length of neck and base zones (Fig. 1a). CD36 is most abundant in endothelial cells, ECs

(Fig. 1b). CD36 is also present exclusively on parietal cells among gastric epithelial cells as documented by laser captured micro-dissected cells and using mice with lineage-specific GFP expression in parietal cells[28,29]. Here we show that CD36 localizes to the parietal cell (PC) basolateral side adjacent to ECs and is excluded from the luminal side, identified by apical membrane-associated ezrin (Fig. 1c). CD36 localization on PCs contrasts with its apical lumen-facing localization in small intestinal enterocytes[26,30] and suggests that it does not function in nutrient absorption from the gastric lumen. CD36 is present in ghrelin and gastrin cells (Fig. 1d-e) but absent from chief (GIF staining), enteroendocrine (chromogranin A), foveolar/pit, mucous neck, and tuft (Doublecortin like kinase-1, DCLK1) cells (Supplementary Fig. 1d–f). In summary, in the stomach CD36 is expressed in parietal, endothelial and some enteroendocrine cells. The higher CD36 expression in the corpus as compared to the antrum can be related to the higher vascularization of the corpus and the lack of parietal cells in the antrum. The expression on enteroendocrine cells (gastrin and ghrelin) does not contribute significantly to total CD36 expression levels since these cells are a small percent (1–2%) of total cells in the mucosa.

*Cd36* deletion did not alter gastric mRNA levels of genes specific to PCs; ATP4B, VEGFB, neck cells; Tff1, pit cells; mucin (Muc5AC) or the enteroendocrine cell marker chromogranin A (ChA) (Supplementary Fig. 1g). The mRNA for gastrin was higher, those for somatostatin and ghrelin lower (Fig. 1f–h). The mRNA for leptin was almost completely (94%) suppressed (Fig. 1i). Plasma gastrin levels were higher at 4 h refeeding (Fig. 1j) in *Cd36*−/− mice in parallel to tissue mRNA and in line with the stomach being the main source of gastrin.

In contrast to the constitutive and slow leptin secretion by adipose tissue[5,31], gastric leptin is released by chief cells[32,33] in a rapidly regulated secretion pathway sensitive to vagal input and raises circulating leptin after hormone stimulation (pentagastrin and cholecystokinin) or food intake[6,32–34]. Plasma leptin in fasted *Cd36*−/− and WT mice did not differ, but leptin increase at 4 h after refeeding was blunted in *Cd36*−/− mice (Fig. 1k). We examined if this reflects the absence of the established role of CD36 in mediating orosensory fatty acid perception by mouth taste buds, transmitted via the gustatory nerve. This perception is integral to vagal regulation of early cephalic responses to food intake[35–37]. Cephalic responses were determined at 15 min after an oral fat meal in fasted WT and *Cd36*−/− mice. Leptin levels increased in WT mice and the increase was suppressed in *Cd36*−/− mice (Fig. 2a). Nongastric responses such as plasma insulin (Fig. 2b) and pancreatic polypeptide (Fig. 2c) were almost absent in *Cd36*−/− mice. Stomach acetylcholine was reduced in *Cd36*−/− mice (Fig. 2d). These data indicated that impaired orosensory FA sensing reduces vagal input after food intake and consequently the gastric secretion of leptin.

Acid secretion by PCs is upregulated by gastrin and acetylcholine and inhibited by somatostatin (SST)[38]. Gastrin stimulates ECL cell release of the secretagogue histamine and potentiates cholinergic action in PCs. Surprisingly, gastric acid secretion, examined using pylorus-ligation before and after the cholinomimetic carbachol or histamine, was similar for WT and *Cd36*−/− mice (Fig. 2e). However, carbachol-induced secretion of gastric intrinsic factor (GIF) by chief cells was reduced in *Cd36*−/− stomachs (Fig. 2f) although GIF expression was unchanged (Fig. 2g). Expression of M1 muscarinic receptors, primarily found in chief cells[39] was higher (Fig. 2h), possibly a compensatory effect to the low GIF output, while expression of M3 and M4, involved in PC acid secretion[40] was unchanged. Thus, *Cd36*−/− stomachs do not display defective carbachol stimulation of acid secretion while carbachol stimulated GIF release is reduced, suggesting selective muscarinic receptor impairment.

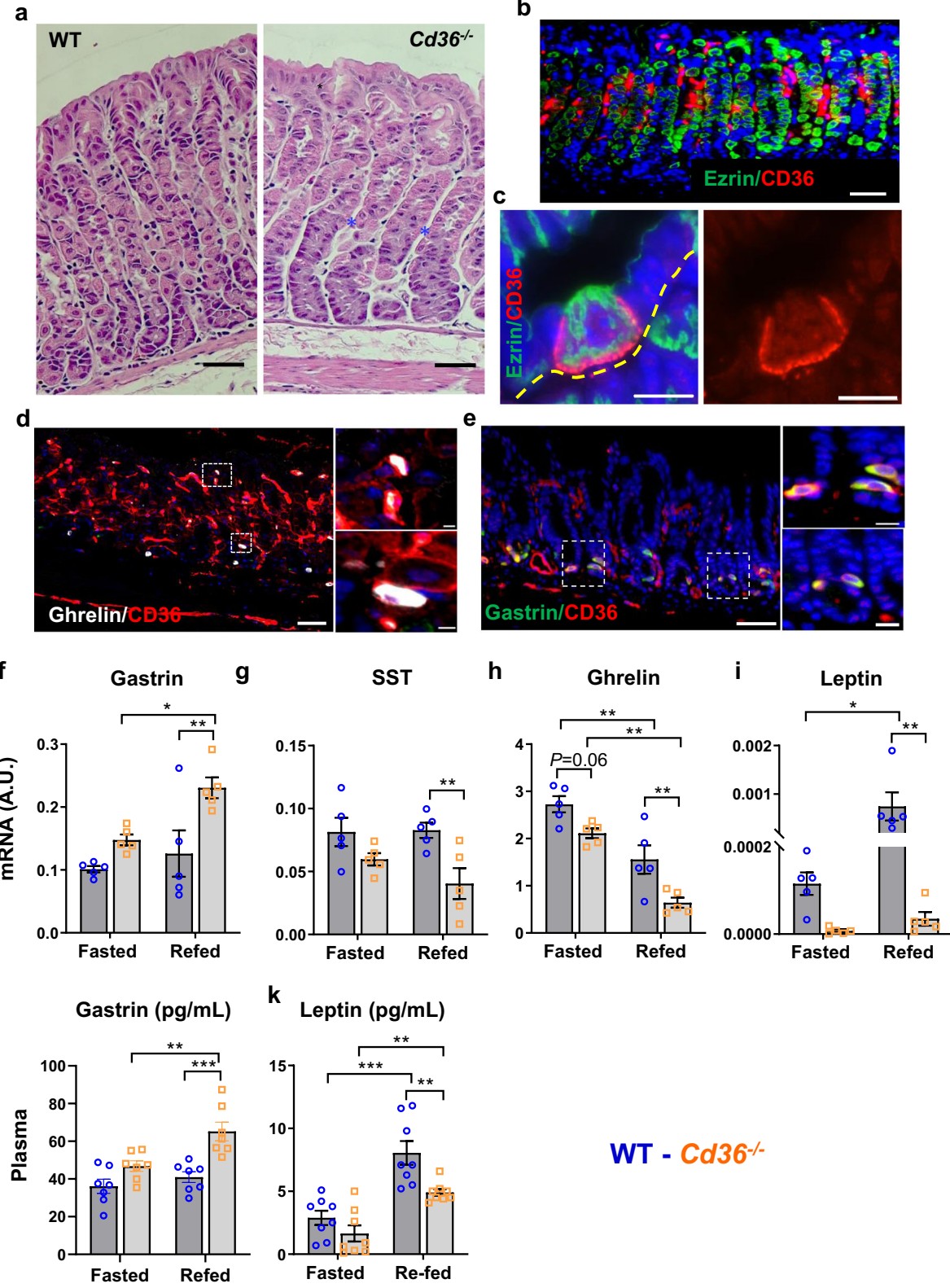

Corpus morphology by transmission electron microscopy (TEM) showed in *Cd36*−/− tissue altered interface between PCs and endothelial cells (Fig. 3a, blue asterisks and white arrows), with *Cd36*−/− PCs harboring abundant mitochondria within the normally mitochondria-free region of interface between the PCs and ECs. This area normally accumulates VEGFB, suggesting it is critical for PC communication with endothelial cells and exchange of metabolites with the blood[29]. Extracellular matrix (ECM) remodeling was evident under steady state conditions in stomachs from nonfasted *Cd36*−/− mice; specifically higher mRNA for fibronectin (Fig. 3b) and fibronectin deposition around vessels and in submucosa (Fig. 3c). Other ECM markers, including laminin and collagen 1α did not change (Fig. 3b). Inflammation markers assessed during fasting and refeeding

**Fig. 1 CD36 is abundant in the corpus on endothelial and parietal cells (PCs) and *Cd36* deletion alters markers of gastric function and vagal input. a** Representative histology and immunohistochemistry of a stomach from a wildtype (WT) mouse (left) and a $Cd36^{-/-}$ mouse (right) showing the altered organization of acid-producing parietal cells (PC) around blood vessels in gastric glands (blue asterisks). **b** CD36 expression in the corpus is most abundant on endothelial cells and is also detected in PCs. **c** In PCs, CD36 (red) is at the basolateral membrane (dotted yellow line) and is excluded from the apical membrane marked by ezrin (magenta) and in contact with the lumen (blue dotted line). **d**, **e** Immunostaining for ghrelin and gastrin (right insets: magnification of cell markers colocalized with CD36). Scale bar: 50 μm, except for **c** and inserts in **d**, **e** (10 μm). **b–e** are stomachs from WT mice. Scale bar: 50 μm. **f–i** mRNA for hormones in fasted or 4 h refed mice. All mRNA adjusted to 36b4. **j** Plasma gastrin and **k** plasma leptin in fasted or 4 h refed mice. SST somatostatin. Significance was calculated using a two-way ANOVA followed by *post hoc* tests by Sidak multiple comparisons. *$P < 0.05$, **$P < 0.01$, ***$P < 0.001$. Data are means ± SEM, **f–i**: $n = 5$, **j** $n = 7$, **k** $n = 8$.

showed higher expression in $Cd36^{-/-}$ tissue; IL6, the chemokine MIP-2, which regulates leukocyte recruitment, and the neutrophil marker S100A8 were increased while anti-inflammatory IL10 trended lower (Fig. 3d). These data suggest compromised gastric tissue in $Cd36^{-/-}$ mice.

**Epithelial renewal after injury is impaired in stomachs from $Cd36^{-/-}$ mice.** The gastric epithelium can experience injury from pathogens, alcohol or drugs, and PC renewal is critical to the tissue's repair ability. The presence of ECM remodeling and inflammation in the $Cd36^{-/-}$ stomach prompted us to examine its response to stress. We used the acute injury model of high-dose tamoxifen (TAM), which causes PC-targeted, estrogen-independent toxicity that kills nearly all PCs[41,42]. As this is followed by rapid de novo PC renewal, progenitor proliferation and differentiation into gastric units can be evaluated. Lineage commitment can also be assessed from measuring the fraction of proliferating chief cells expressing neck cell markers, such as Griffonia Simplicifolia lectin II (GSII), referred to as SPEM cells[43]. Mice were TAM injected for 3 days and gastric recovery evaluated 5 days later. As expected[41,42], TAM induced PC atrophy in WT and $Cd36^{-/-}$ mice to the same extent and increased cell proliferation throughout the gastric glands in both groups (Supplementary Fig. 2a–c). However, gland recovery differed between genotypes, while at day 5 post-TAM, proliferation (Fig. 4a, d) and PC atrophy (Fig. 4b, e) were largely reversed in WT mice, in $Cd36^{-/-}$ mice although proliferation was reversed, PC renewal was 41% lower than WT mice. This indicated suboptimal progenitor differentiation to PC, and the new glands were abnormally elongated (Fig. 4f). Consistent with a differentiation defect, the fraction of chief cells that had undergone palingenosis to the metaplastic, proliferative SPEM lineage (SPEM cells co-express neck and chief cell markers; GSII[+]/GIF[+]) was 60% higher in the $Cd36^{-/-}$ relative to the WT corpus (Fig. 4g). The lower PC renewal and increase in SPEM cells indicate that defects in progenitor differentiation to PCs and in lineage commitment drive the impairment of mucosal renewal.

**$Cd36$ deletion reduces stomach FA uptake and mucosal respiration.** FA oxidation is particularly critical for tissue stem cells[18,44] and reduced FA oxidation impairs stem cell functionality and the ability for optimal differentiation[45]. We examined the effect of $Cd36$ deletion on gastric FA metabolism. Deletion of $Cd36$ reduces FA uptake in heart, skeletal muscle, and fat tissue but reduction is not observed in other tissues such as liver or kidney[46]. To our knowledge, uptake of long-chain FA by the stomach has not been examined whether in the presence or absence of CD36. Mice intravenously injected with [3H]oleic showed robust oleic acid uptake by the corpus and uptake was significantly reduced in $Cd36^{-/-}$ mice (Fig. 5a). We next determined if the reduced uptake alters tissue oxygen consumption by corpus mucosa from WT and $Cd36^{-/-}$ mice. The $Cd36^{-/-}$ mucosa consumed 26% less oxygen in the oxidative

phosphorylative state (OxPhos C-I and C-II) and had lower maximal respiratory flux (ETS) (Fig. 5b). Mitochondrial coupling efficiency (of oxidation and ATP synthesis, RCR) trended lower in $Cd36^{-/-}$ compared to WT mucosa ($P = 0.07$, Fig. 5c). Mitochondrial FA oxidation was determined by mass spectrometry of oxidation intermediates. The $Cd36^{-/-}$ stomachs had lower levels of short-chain acylcarnitines but accumulated long-chain acylcarnitines, a pattern indicative of inefficient mitochondrial FA oxidation[47,48] (Fig. 5d). In addition, cardiolipin, an essential constituent of the inner mitochondrial membrane, increased in refed $Cd36^{-/-}$ stomachs and had higher polyunsaturated FA content (Fig. 5e, f), a remodeling similar to that described in aging and early stages of diabetes[49,50]. These functional abnormalities associated with morphological changes in mitochondria (Fig. 5g, h), which were more circular with a reduced aspect ratio (major/minor axis, Fig. 5h). Together these studies show that the corpus relies on circulating FA, and chronically reducing FA supply reduces mitochondrial efficiency, and diminishes tissue respiration and mitochondrial FA oxidation, which impairs PC progenitor ability for optimal differentiation to PCs.

**The $Cd36^{-/-}$ stomach is depleted of TAG and has higher phospholipids.** Untargeted lipidomic analyses showed that stomachs from fasted $Cd36^{-/-}$ mice had depleted triglycerides (TAG) (Fig. 6a) whether diet-derived or de novo synthesized (Supplementary Fig. 3a) while phospholipids are increased. Refeeding increased TAG and diacylglycerols (DAG) in WT stomachs, but most TAGs remained reduced in $Cd36^{-/-}$ stomachs (Fig. 6a, b) while phosphatidylcholine (PC), phosphatidylethanolamine (PE) (Fig. 6a, c), and phosphatidylglycerol (PG) increased (Fig. 6a). Increase of the percent phospholipid has been reported in chronic atrophic gastritis and gastric ulcers[51]. In addition, there was the remodeling of PC and PE polyunsaturated FA (PUFA) with higher percent arachidonic acid (ω6 20:4) (Fig. 6d) and ratio of 20:4 to ω3 FA (20:5, 22:5 and 22:6). Consistent with lack of TAG/DAG response to refeeding, $Cd36^{-/-}$ mice had lower mRNA of a key TAG synthesis enzyme GPAT4 and failed to increase mRNA levels of AGPAT1 after refeeding, while adipose triglyceride lipase (ATGL), the key to FA mobilization, increased during fasting in line with depletion of TAG stores (Fig. 6e, f). Expression of glycolytic enzymes was unchanged, but pyruvic and lactic acids (Supplementary Fig. 3b) and Krebs cycle fumarate and malate (Supplementary Fig. 3c) were reduced. Thus, CD36 deletion remodels gastric tissue lipids as TAG are depleted and phospholipids increase, the latter change reminiscent of trends reported in gastritis and ulcers[51].

**Deletion of $Cd36$ in parietal cells does not alter the response to injury.** The impaired recovery from injury in $Cd36^{-/-}$ stomachs reflected dysfunction of PC progenitor differentiation. We examined if CD36 absence in mature PCs can drive the defect in epithelial renewal. We generated a mouse model with PC-specific

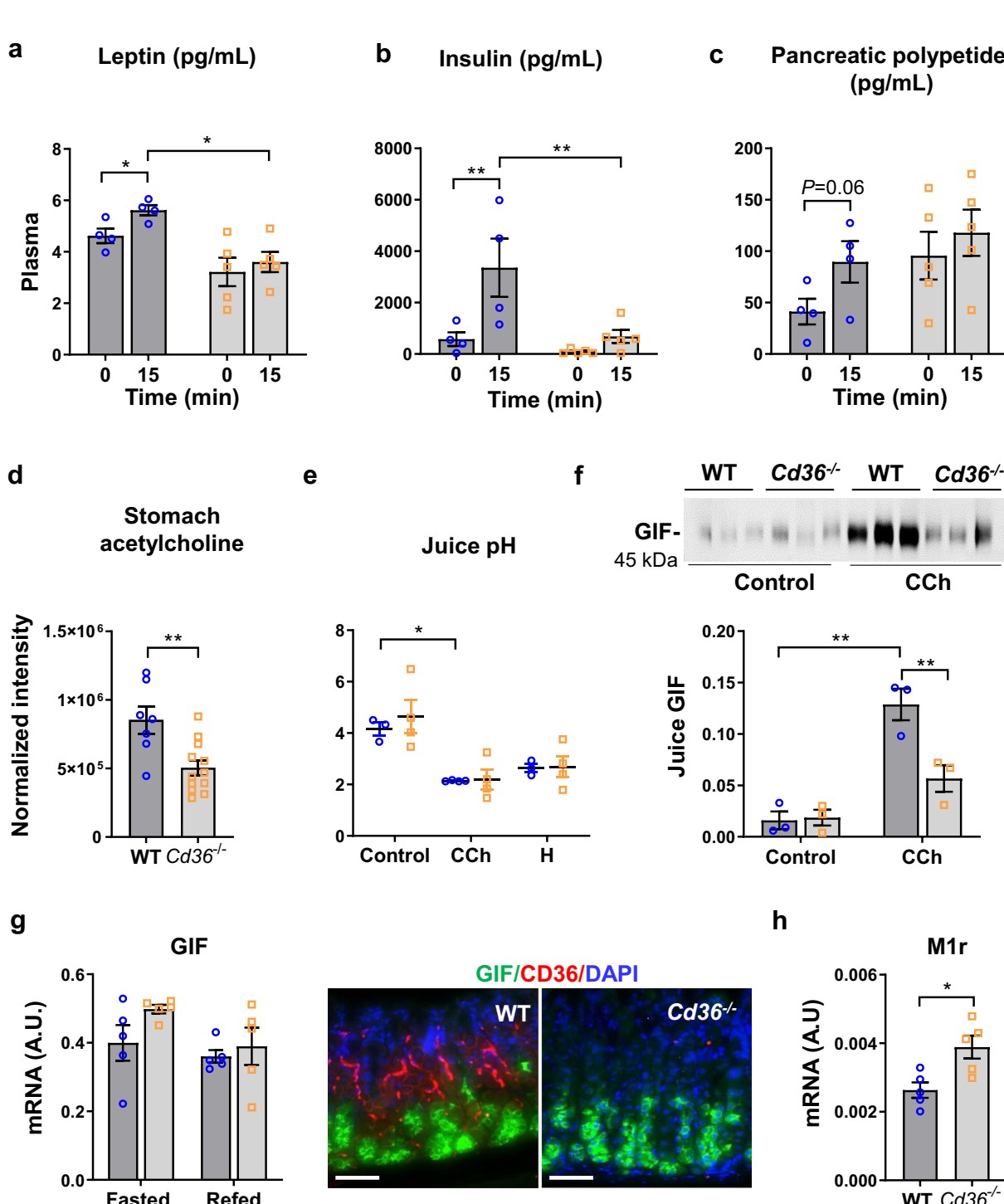

**Fig. 2 Cephalic response to oral high fat and gastric secretions in germline $Cd36^{-/-}$ mice. a–c** Impaired response of plasma leptin, insulin, and pancreatic polypeptide in fasted mice and 15 min after an oral high fat meal. **d** Stomach acetylcholine levels in 4 h refed mice. **e,f** Gastric secretions at 1 h after pylorus ligation in 18 h fasted mice given i.p. saline (Control), carbachol (CCh, 60 μg/kg) or histamine (H, 10 mg/kg). **e** Gastric juice pH. **f** Immunoblots of gastric juice; quantification of gastric intrinsic factor (GIF)/total protein in juice showing reduced GIF output in $Cd36^{-/-}$ mice given carbachol. **g** Chief cells markers by qPCR and immunostaining (green) in corpus of wildtype (WT) and $Cd36^{-/-}$ mice, scale: 50 μm. **h** mRNA of muscarinic receptor 1 (M1r) in the corpus of WT and $Cd36^{-/-}$ mice. Significance was calculated using a two-way ANOVA followed by post hoc tests by Sidak multiple comparisons or a two-sided unpaired $t$ test (**d** and **h**). *$P < 0.05$, **$P < 0.01$. **a–c** $n = 4–5$ per condition/group; **d**: $n = 7–12$; **e,f** $n = 3–4$; **g,h** $n = 5$.

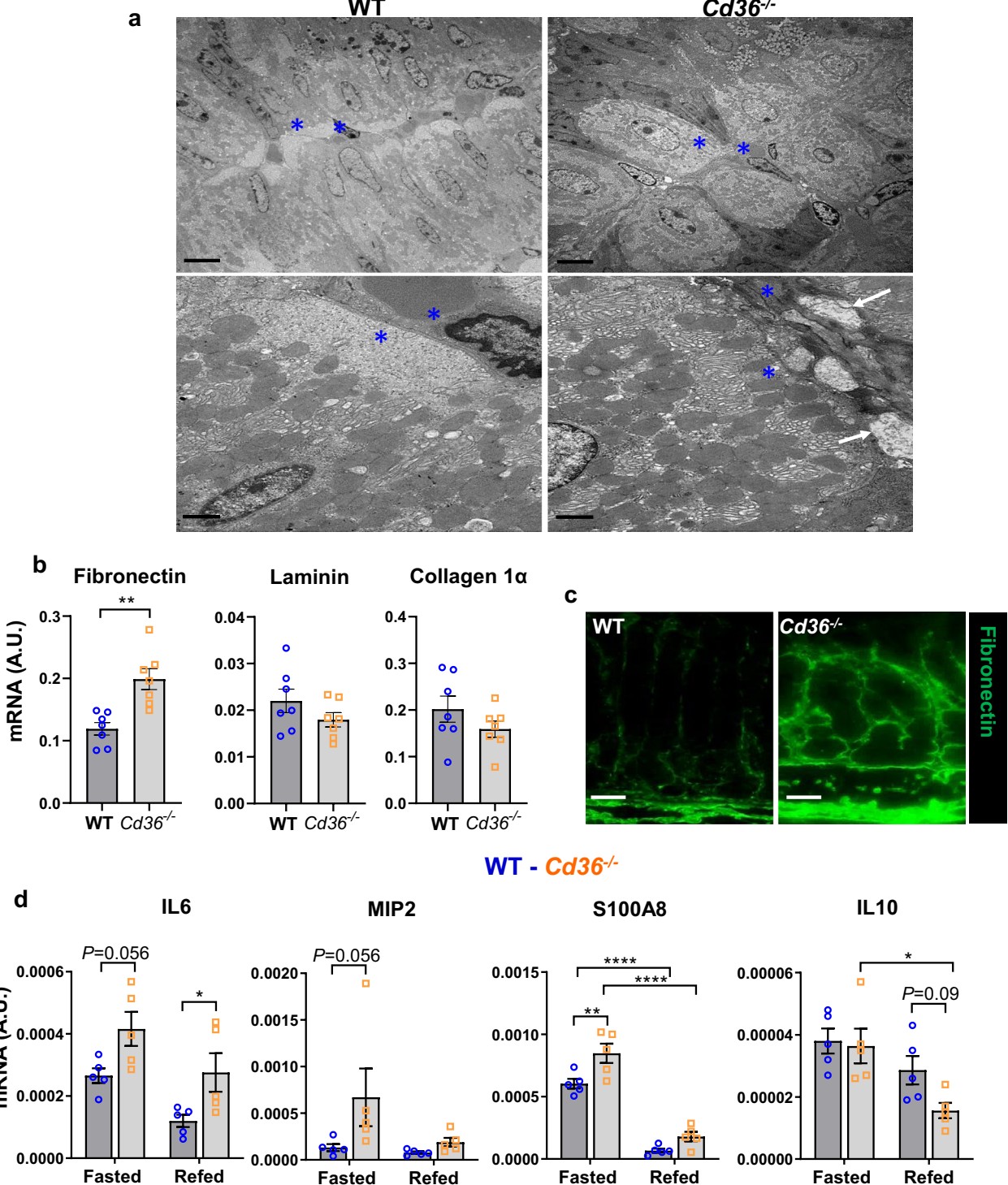

**Fig. 3 Increase in extracellular matrix (ECM) deposition and inflammation markers in *Cd36*⁻/⁻ stomachs. a** Transmission electron microscopy (TEM) of parietal cells in the *Cd36*⁻/⁻ corpus show a lack of mitochondria-free area at the basolateral side (blue asterisks) and ECM deposits between the PC membrane and blood vessels (white arrows). Scale bars: 6 μm (top) and 1 μm (bottom) panels. **b** mRNA of ECM proteins fibronectin, laminin and collagen1α in nonfasted mice. **c** Immunostaining of fibronectin (green) in corpus of WT and *Cd36*⁻/⁻ mice. Scale bar: 50 μm. **d** mRNA of inflammation and neutrophil markers. Significance was calculated using a two-way ANOVA followed by post hoc tests by Sidak multiple comparisons. *$P < 0.05$, **$P < 0.01$, ***$P < 0.001$. Data are means ± SEM. **b** $n = 7$, **d** $n = 5$.

*Cd36* deletion (PC-*Cd36*⁻/⁻) (Supplementary Fig. 4a). The *Cd36*⁻/⁻ PCs, like PCs of *Cd36*⁻/⁻ mice, showed mitochondrial migration toward adjacent capillaries but no alterations in the plasma membrane or in the appearance of mitochondria (Supplementary Fig. 4b, c). The corpus did not show altered

expression of markers for PCs, gastrin, or genes related to ECM remodeling or inflammation (Supplementary Fig. 4d, e). Unlike in *Cd36*⁻/⁻ mice, oleate uptake by the corpus was unchanged in PC-*Cd36*⁻/⁻ mice (Supplementary Fig. 4f). The PC-*Cd36*⁻/⁻ stomachs recovered from TAM injury like those of floxed

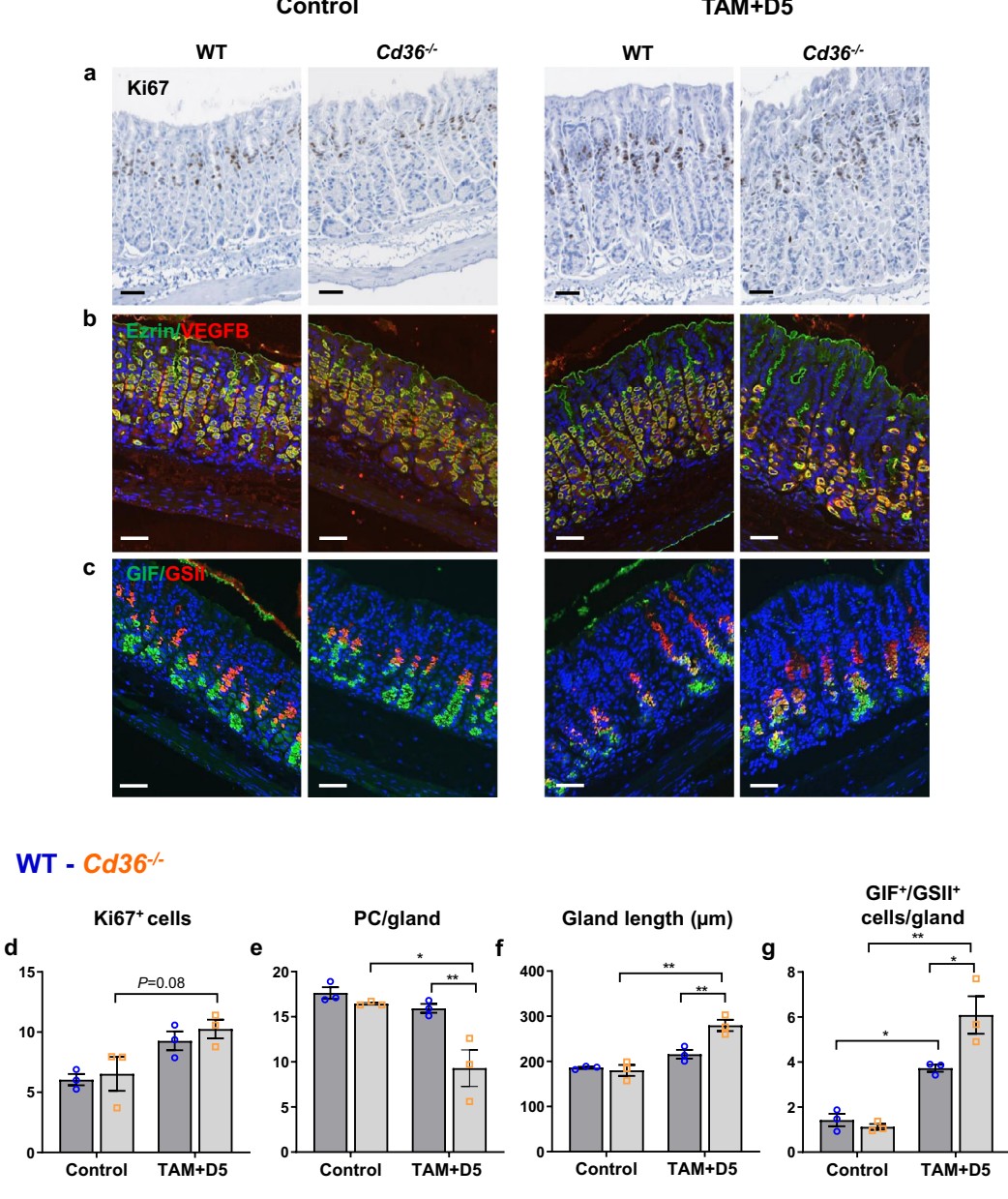

**Fig. 4 CD36 is required for parietal cell (PC) renewal and recovery during gastric injury. a–c** Representative histology and immunohistochemistry of wildtype (WT) and $Cd36^{-/-}$ mice that were either untreated (control), or injected with TAM for 3 days followed by 5 days of recovery (TAM + D5); **a** proliferation marker ki67; **b** vascular endothelial growth factor B (VEGFB), red; ezrin, green; **c** GIF, green; GSII, red. Scale bar: 50 μm. Quantification of **d** ki67+ cells and **e** parietal cells per gastric unit, **f** gland length, and **g** GIF/GSII+ cells per gastric unit (yellow). Significance was calculated using a two-way ANOVA followed by *post hoc* tests by Sidak multiple comparisons. *$P < 0.05$, **$P < 0.01$. Data are means ± SEM, $n = 3$ mice/group.

controls (Supplementary Fig. 5a–d). These data show that while global $Cd36$ deficiency results in dysfunctional progenitors and reduced PC renewal, deletion of $Cd36$ in mature PC is not, by itself, sufficient to cause the corpus abnormalities and delayed injury recovery seen in $Cd36^{-/-}$ stomachs.

**$Cd36$ deletion in endothelial cells reduces stomach FA uptake and recapitulates major gastric abnormalities of $Cd36^{-/-}$ mice.** Recent findings documented a primary role of endothelial cell CD36 in FA uptake by the heart, muscle, and brown adipose tissue[46]. As in these tissues, CD36 in the stomach is most robustly expressed on ECs in the corpus. We examined if EC-$Cd36^{-/-}$ mice can recapitulate the morphologic and metabolic alterations and the delayed PC recovery observed in $Cd36^{-/-}$ stomachs.

Similar to $Cd36^{-/-}$ mice and unlike PC-$Cd36^{-/-}$ mice (Supplementary Fig. 4f), the EC-$Cd36^{-/-}$ mice (Fig. 7b) had reduced corpus uptake of injected [³H]oleate. The EC-$Cd36^{-/-}$ stomachs displayed many of the abnormalities seen in $Cd36^{-/-}$ mice; mitochondrial migration towards capillaries in PCs, remodeling of mitochondrial cardiolipin with enrichment in ω6 PUFA, less elongated mitochondria with a reduced aspect ratio (Fig. 7c–e) and increases in MCP1 and fibronectin (Fig. 7f, g).

Untargeted metabolomic-lipidomic analyses revealed that EC-$Cd36^{-/-}$ stomachs (Supplementary Fig. 6a–b) like $Cd36^{-/-}$ stomachs had reduced diet-derived TAG although de novo TAG species increased, reflecting more glucose utilization for lipogenesis, likely due to CD36 presence in PCs (Fig. 7a). As in $Cd36^{-/-}$ mice, PC and PE had more percent 20:4 and higher 20:4 to ω3-PUFA ratios (Supplementary Fig. 6d).

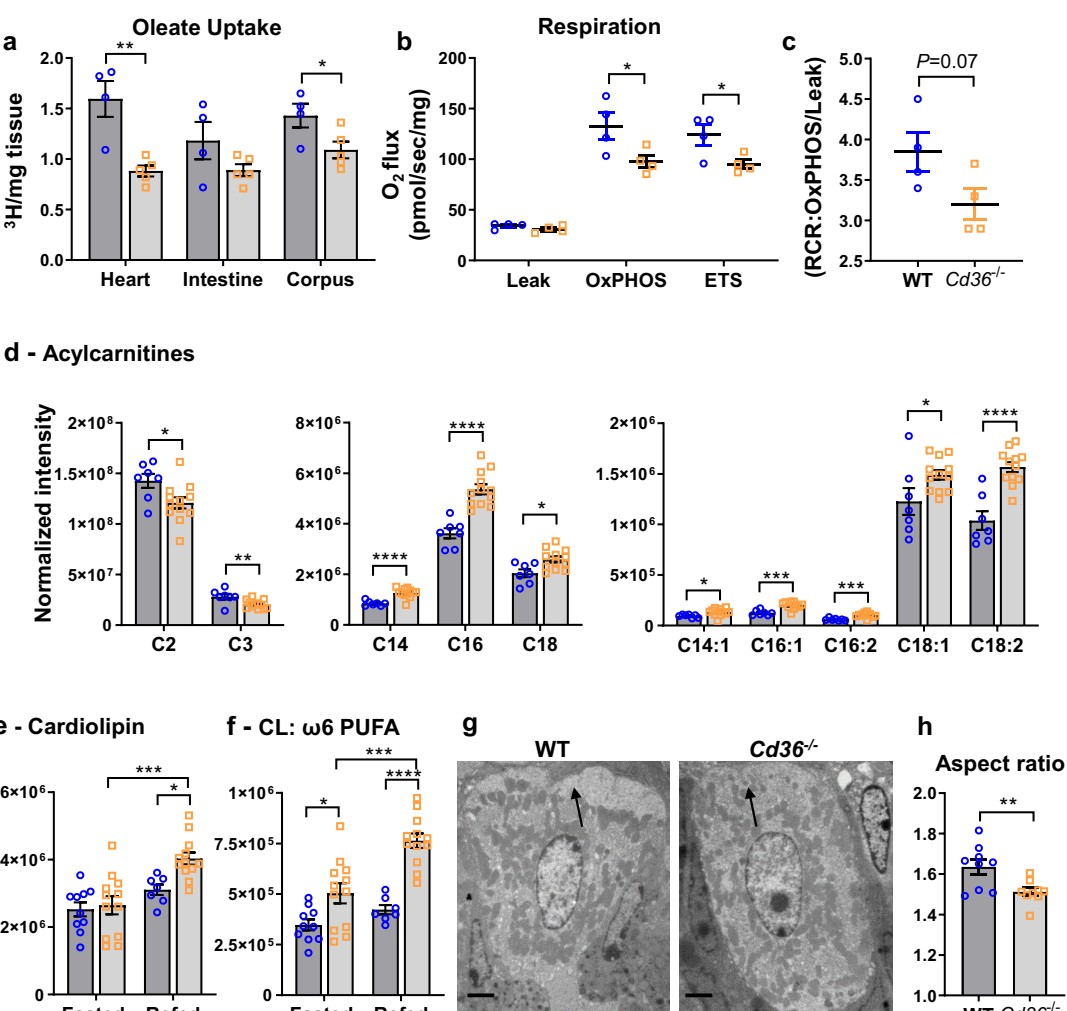

**Fig. 5 Decreased tissue FA uptake and respiration by *Cd36*^−/−^ stomachs. a** Tissue uptake of oleic acid in wildtype (WT) and *Cd36*^−/−^ mice. Four-month-old mice 16 h fasted mice, were given a retro-orbital injection of [³H]oleic acid and tissues collected 5 min later for measuring radioactivity. **b** High-resolution respirometry (Oxygraph-2k) of permeabilized corpus mucosa from WT and *Cd36*^−/−^ mice. Leak: plus malate (1 mM), glutamate (10 mM) and pyruvate (5 mM). OxPHOS: Oxygen consumption, plus ADP (5 mM) and succinate (10 mM). ETS: Maximum flux after FCCP uncoupling. **c** Ratio of OxPHOS/leak in WT and *Cd36*^−/−^ mucosa showing trend for lower coupling efficiency. **d** *Cd36*^−/−^ stomachs have lower levels of short-chain acylcarnitines but accumulate long-chain acylcarnitines. **e, f** cardiolipin (CL) enrichment with ω6 PUFA. **g** transmission electron microscopy (TEM) showing altered morphology of PC mitochondria (arrow) in *Cd36*^−/−^ mice, scale bar: 2 µm. **h** Mitochondrial circularity and aspect ratio (major/minor axis). Significance was calculated using a two-sided unpaired *t* test or a two-way ANOVA followed by *post hoc* tests by Sidak multiple comparisons (**e, f**, and **h**). *$P < 0.05$, *$P < 0.01$, ***$P < 0.001$, ****$P < 0.0001$. Data are means $\pm$ SEM. **a** $n = 4$–5; **b**–**c** $n = 4$; **d**–**f** $n = 7$–12; **h** $n = 600$ mitochondria from 9 PCs/genotype.

However, unlike the *Cd36*^−/−^ mice which survived the TAM protocol, the EC-*Cd36*^−/−^ mice in two separate experiments died during the recovery phase. The reason for the mortality is unclear and suggests unresolved inflammation from the combination of gastric damage after TAM with damage from *Cd36* deletion in endothelial cells.

**Low CD36 expression associates with stomach disorders in humans**. The current findings suggest that CD36 deficiency impairs gastric recovery from injury that would predispose to pathology, so we explored their potential human relevance. PrediXcan analysis, which estimates the genetic component of gene expression using models trained with reference transcriptome data

(v8) from the Genotype-Tissue Expression (GTEx) Consortium[52,53], was applied to Vanderbilt University's BioVU genomic resource where electronic health records (EHR) are tied to whole-genome genetic information[54]. We identified significant associations between genetically reduced CD36 expression and increased incidence of gastric ulcer, gastritis, duodenitis, and nonspecific hemorrhage of the gastrointestinal tract (Table 1). These findings support the functional data in mice and further suggest that CD36 is also likely important for the maintenance of the gut epithelium. In line with this, exploration of the UK Biobank, another large-scale EHR database that combines in-depth genetic and health information from half a million UK participants[55] identified a CD36 SNP rs144921258 that strongly associates ($p = 7.9e$-17) with gastrointestinal hemorrhage as the

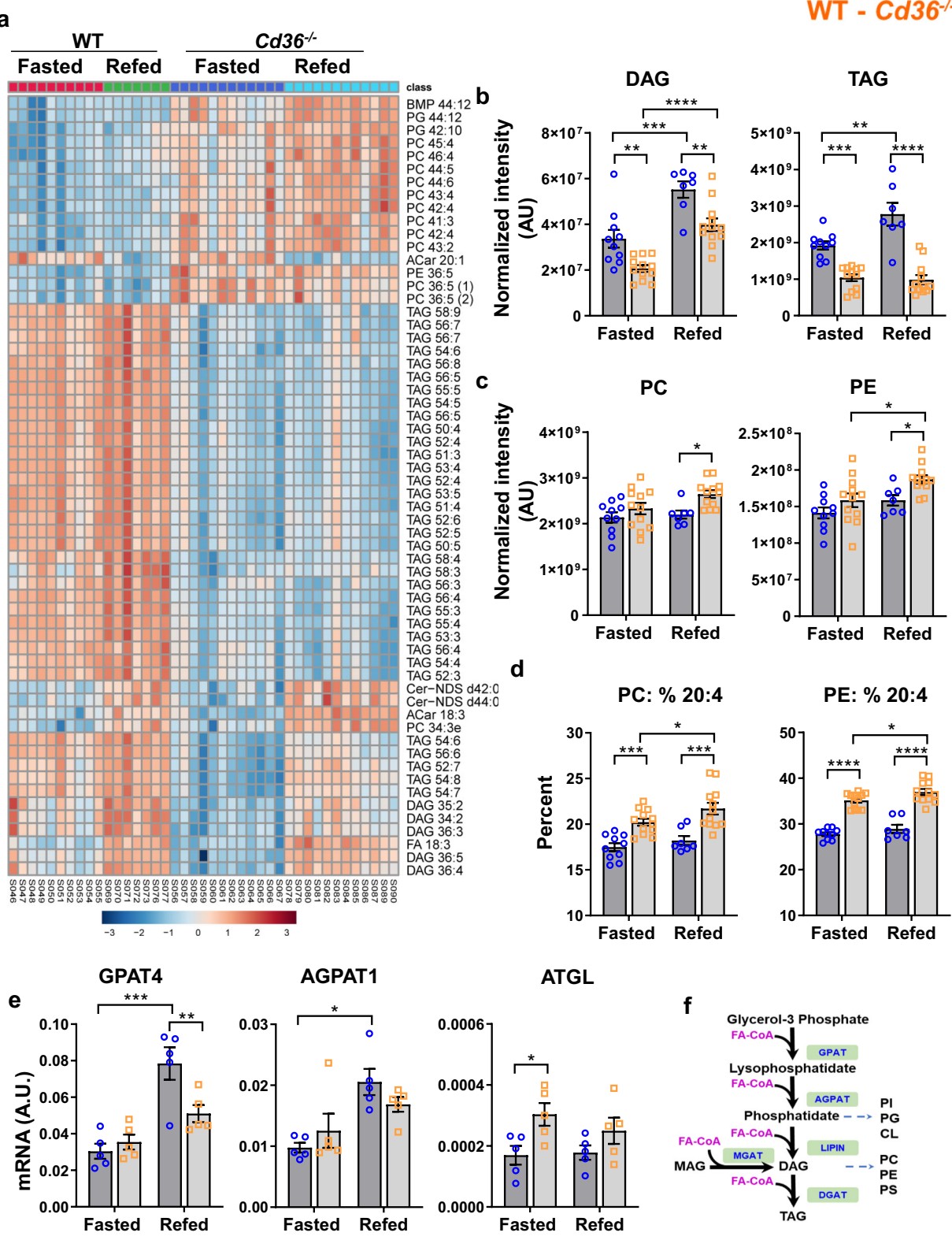

underlying primary cause of death (Fig. 8a). The SNP's top association is this phenotype across 1403 binary phenotypes tested. The SNP occurs primarily in individuals of European ancestry with 0.15 % minor allele frequency. It lies in intron 1 close to the major 1B promoter of CD36 (Fig. 8b) where it is predicted to disrupt an enhancer element based on H3K4me1 CHiP-Seq data from Roadmap Epigenomics Consortium[56]. The gene area where the SNP localizes is rich in epigenetic signatures (H3K4me1, H3K4me2, H3K4me3, H3K27ac) and methylation sites[57]. Fig. 8b shows the position of the SNP relative to those previously associated to methylation sites (CpG) and to reductions of CD36 expression and lipid handling[57].

**Fig. 6 Decreased lipid storage and phospholipid remodeling in _Cd36_<sup>−/−</sup> stomach. a** Heat map of top 60 metabolites/lipids in fasted and re-fed wildtype (WT) and _Cd36_<sup>−/−</sup> stomach. **b** The sum of identified diacylglycerol (DAG) and triacylglycerol (TAG) species was reduced in _Cd36_<sup>−/−</sup> stomachs. **c** _Cd36_ deficient stomachs accumulate more phospholipids with **d** higher % of 20:4. **e** _Cd36_<sup>−/−</sup> mice have lower mRNA levels of TAG synthesis enzyme glycerol-3-phosphate acyltransferase 4 (GPAT4) but increased levels of the TAG lipolytic enzyme adipose triglyceride lipase (ATGL). **f** Schematic summary of key steps in TAG synthesis with intermediates that serve as precursors of other lipid species, such as phospholipids (PL). CL cardiolipin, FA fatty acids, MAG monoacylglycerol, PC phosphatidylcholine, PE phosphatidylethanolamine, PG phosphatidylglycerol, PI, phosphatidylinositol, PS, phosphatidylserine, PUFA, polyunsaturated FA. Significance was calculated using a two-way ANOVA followed by post hoc tests by Sidak multiple comparisons. *$P < 0.05$, **$P < 0.01$, ***$P < 0.001$, ****$P < 0.0001$. Data are means ± SEM, $n = 7$–12 mice/group for **a**–**d**, and $n = 5$ for **e**.

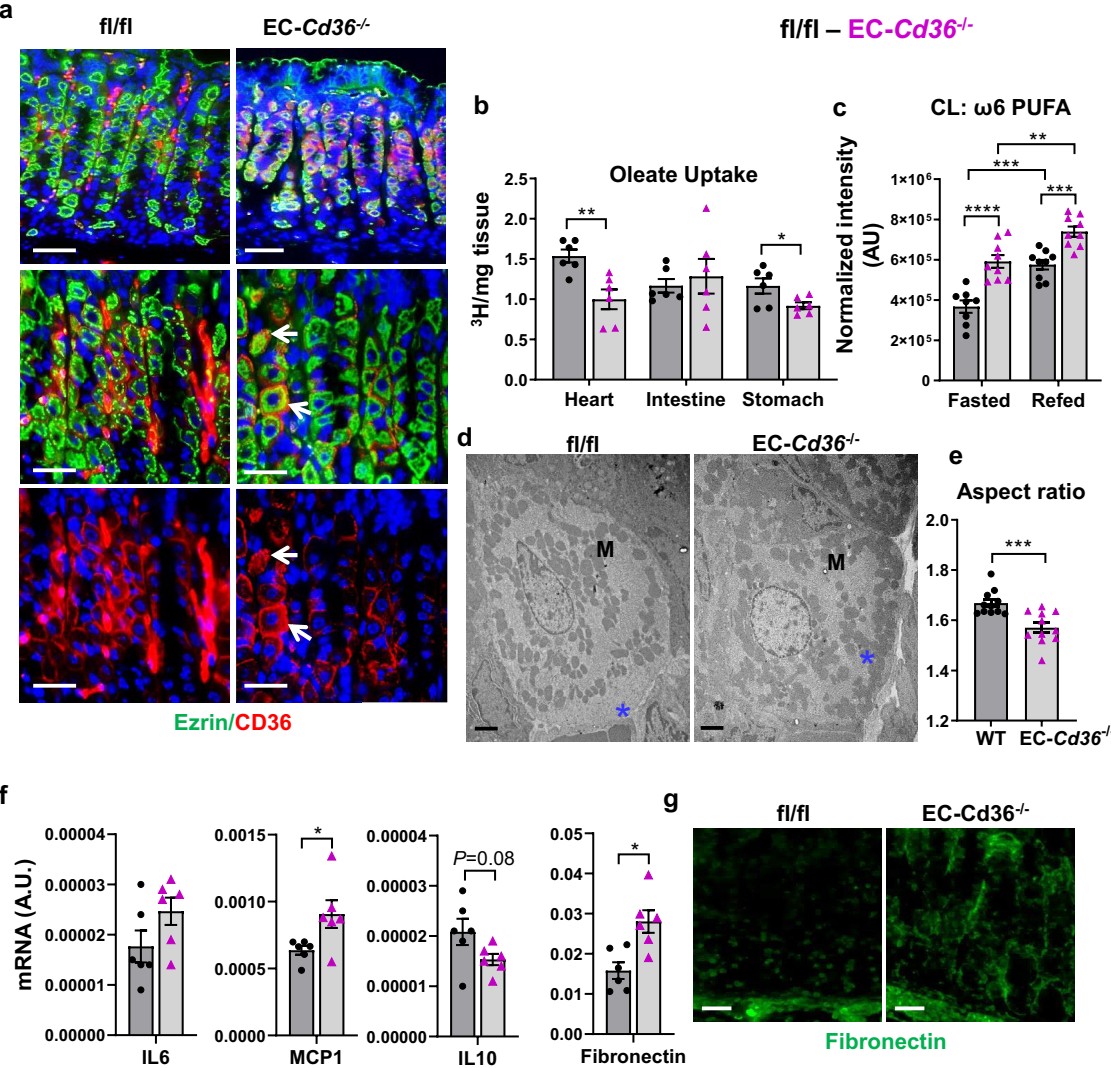

**Fig. 7 Deletion of _Cd36_ in endothelial cells (EC) reduces corpus fatty acid uptake and recapitulates morphological abnormalities of germline _Cd36_<sup>−/−</sup> mice. a** Gastric sections from floxed controls (left panels) and EC-_Cd36_<sup>-/-</sup> mice (right panels) showing absence of CD36 expression in the gastric endothelium and CD36 expression in parietal cells (PCs) (white arrows, bottom right). Scale bars: 50 μm (top) and 20 μm (bottom) panels. **b** Corpus uptake of oleic acid is reduced in EC-_Cd36_<sup>−/−</sup> mice. 16 h fasted 4-month-old mice were given a retro-orbital injection of [<sup>3</sup>H]oleic acid and tissues collected for measuring radioactivity 5 min later. **c** Sum of cardiolipin species enriched with ω6 PUFA **d** Transmission EM showing EC-_Cd36_<sup>−/−</sup> PCs lack the mitochondria ("M")-free area near the plasma membrane bordering capillaries (right panel). Scale bar: 2 μm. **e** Quantification of mitochondrial circularity and aspect ratio (major/minor axis). **f** mRNA expression of gastric hormones, fibronectin, and inflammatory markers in corpus of EC- _Cd36_<sup>−/−</sup> mice. **g** Fibronectin staining (green) in corpus of control and EC-_Cd36_<sup>−/−</sup> mice, scale bar: 50 μm. Significance was calculated using a two-sided unpaired _t_ test or a two-way ANOVA followed by _post hoc_ tests by Sidak multiple comparisons (**c**). *$P < 0.05$, **$P < 0.01$, ***$P < 0.001$, ****$P < 0.0001$. Data are means ± SEM. **b** $n = 6$; **c** $n = 8$–9; **e** $n = 1500$ mitochondria from 11 PCs/genotype; **f** $n = 6$.

## Discussion

Little attention has been paid to understanding the metabolism of the gastric mucosa although it could shed light on the etiology of common gastric diseases. In this study, metabolic and functional findings in three mice models of _Cd36_ deficiency provide strong support for the critical role of CD36 in maintaining optimal gastric tissue function and the ability for self-renewal postinjury (Fig. 9). The major contributions of the work are the following.

**Table 1 Low CD36 expression associates with susceptibility to gastric disorders.**

| Source | *P* value | gene | trait | Number of cases | Number of controls |
|---|---|---|---|---|---|
| BioVU | 0.024 | CD36 | Gastric ulcer | 184 | 21823 |
| BioVU | 0.04 | CD36 | Gastritis and duodenitis | 744 | 19809 |
| BioVU | 0.036 | CD36 | Other specified gastritis | 147 | 19809 |
| BioVU | 0.0088 | CD36 | Gastrointestinal Hemorrhage | 1606 | 55179 |
| UK Biobank | 7.9e-17 | CD36 | GI Hemorrhage as primary cause of death | 5407 | 361194 |

Significant associations are observed between reduced CD36 mRNA and increased incidence of gastric ulcer, gastritis, duodenitis and hemorrhage of the gastrointestinal tract in the Vanderbilt University's BioVU genomic resource. The total number of patients for each trait is the sum of cases and controls for each. Note: There is an overlap of 10 shared cases between gastric ulcer and gastritis duodenitis patients. CD36 SNP rs144921258 found in individuals of European ancestry associates with gastrointestinal hemorrhage as the primary cause of death in the UK Biobank (also see Fig. 8).

a

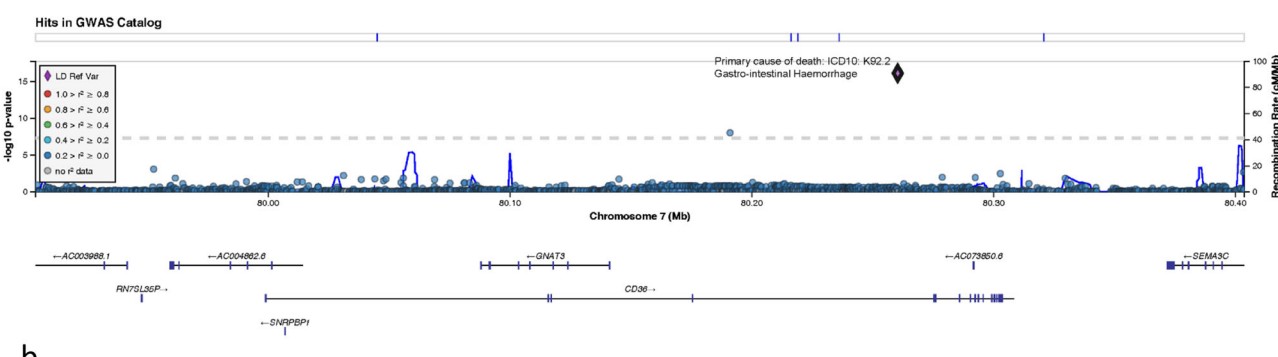

b

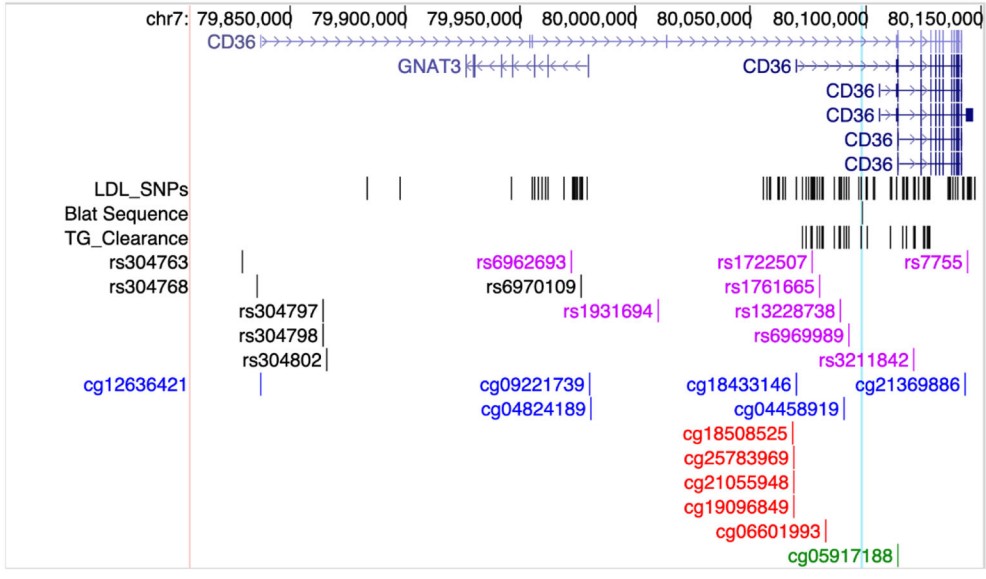

**Fig. 8 Single nucleotide polymorphism rs144921258 associates with primary cause of death from gastrointestinal hemorrhage within the UK Biobank cohort. a** Genomic coordinates along chromosome 7q (x-axis), with negative logarithm of p-values (y-axis) for SNP association with gastrointestinal hemorrhage $p = 7.9^{-17}$. Location of rs144921258 in CD36 on chr7q is indicated by a diamond. The schematic of the *CD36* gene below the plot also depicts the position of GNAT3, which overlaps a distal promoter of CD36. **b** Position of rs144921258 relative to the abundant proximal CD36 promoter 1B, (shown with its long 3′ untranslated region) and to SNPs previously associated with methylation at CpG sites with impact on CD36 expression and lipid handling. Red CpGs: SNP-associated methylation sites that impact CD36 expression. Purple SNPs: SNPs associated with both lipid levels and low CD36 expression. See also Table 1 and Results.

First, we show that Cd36 deletion impairs gastric function, metabolism, and mucosal renewal after injury, through reducing endothelial cell delivery of fatty acids to the corpus and not by damaging parietal cell function. Second, as with other stem cell populations, the long-term defect in FA uptake and oxidation was found to result in less competent progenitor cells that are not amenable to full differentiation into parietal cells, impeding tissue recovery from injury. Finally, the defective release of leptin by the CD36 deficient stomach is an additional potential contributor to the impaired gastric recovery. The clinical impact of our work is

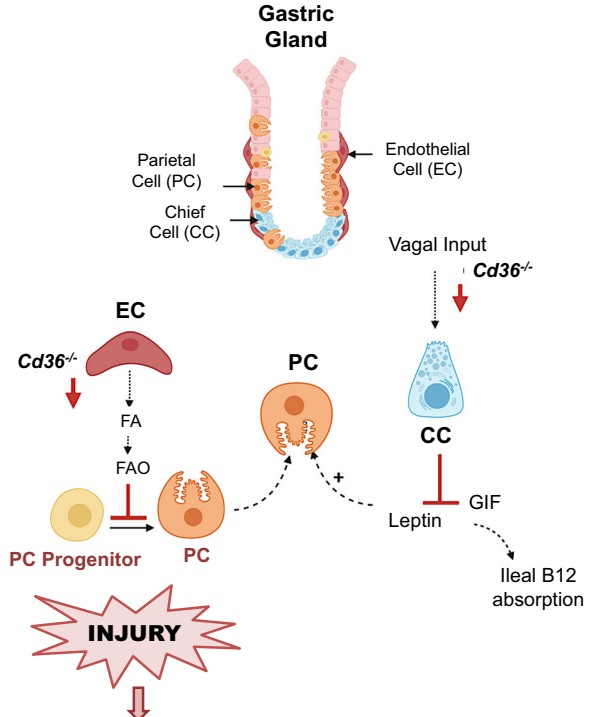

**Fig. 9 CD36 regulates stomach function and tissue renewal postinjury.**
CD36 is abundant in corpus endothelial cells (EC) and expressed in parietal cells (PC). It is not detected in chief cells (CC), but its deficiency through vagal input and likely muscarinic receptor 1, suppresses CC release of leptin and gastric intrinsic factor (GIF), respectively. GIF is necessary for ileal vitamin B12 absorption. CD36 deletion reduces endothelial delivery of fatty acids to the corpus, tissue energy stores and mitochondrial FA oxidation, critical for stem cell functionality. Impaired renewal of the CD36[-/-] gastric epithelium reflects defective PC progenitor differentiation to PCs and not progenitor proliferation or dysfunction of mature PCs. Low gastric leptin secretion could further impair tissue healing. The reduced tissue ability for adequate repair would contribute to the etiology of gastric disease, as supported by data from two biomedical databases (see Fig. 8 and Table 1).

highlighted by the finding that the defect of gastric recovery in low CD36 expression might be one underlying mechanism in the etiology of ulcer, gastritis, and gastrointestinal hemorrhage.

CD36 deficiency altered stomach release of gastrin, GIF, and leptin. In particular suppressed release of GIF which is required for intestinal B12 absorption, chronically could cause B12 deficiency that is unresponsive to oral B12 supplementation. GIF binds B12 in the duodenum and transports it to the ileum where it is absorbed. B12 deficiency causes neuronal demyelination, cognitive impairment, and macrocytic anemia[11,58].

The gastrin increase in CD36 deficiency did not correlate with hypochlorhydria and was probably due to the reduced somatostatin, which inhibits gastrin gene expression and its release[59]. The high gastrin in $Cd36^{-/-}$ stomachs likely sustains acid secretion despite the diminished vagal input[60].

Leptin and ghrelin play a significant role in energy balance, having opposite functions in regulation of hunger and satiety. In the stomach, food intake inhibits ghrelin secretion, while it increases that of leptin[2,3,6,31]. In the present study, CD36 deficiency decreased stomach mRNA levels of both hormones. Leptin secretion usually measured in the circulation reflects adipose

tissue leptin, as the contribution of stomach leptin release is only significant early after a meal. Regulation of leptin secretion by food intake differs between the adipose and stomach tissues, with slow versus rapid responses of adipose and stomach leptin, respectively[5,31]. Thus, immediately after feeding the change in circulating leptin reflects gastric leptin secretion. In CD36 deficiency the marked reduction in leptin secretion consequent to low vagal input could have contributed to the defect in mucosal recovery from injury in the stomach since gastric leptin which reaches the intestine was shown important for intestinal and gastric maintenance[9,61,62].

Uptake of circulating FAs by corpus endothelial cells was found necessary to maintain tissue energy stores and FA oxidation. CD36 upregulates FA oxidation in vivo[63,64] and in vitro through activation of AMPK and PPAR delta[22,65]. FA oxidation is particularly critical for tissue stem cells[18,44] as reduced FA oxidation impairs stem cell functionality[45] and ability for optimal differentiation[15]. In the tamoxifen gastric injury model, we document that CD36 deletion suppresses the conversion of PC progenitors into PCs despite similar progenitor proliferation. Moreover, deeper in the gland, the number of GIF/GSII positive SPEM cells generated by palingenosis from chief cells[66] is increased, and the new glands are often abnormally elongated. Thus, in addition to PC progenitors, both isthmal/neck and chief cell progenitor zones are affected by the loss of CD36.

Unlike $Cd36$ deletion in ECs, its deletion in parietal cells did not reduce corpus FA uptake and did not reproduce the above abnormalities. Thus, the endothelium is a gatekeeper in gastric tissue FA uptake, and CD36 in PCs like CD36 in muscle cells, alone is not sufficient to cause tissue-wide abnormalities[46]. The data are also consistent with the primary role of dysfunctional progenitor differentiation rather than of dysfunctional PCs in the defect in gastric renewal.

The reduced FA uptake in the corpus increased phospholipids and phosphatidylglycerol (PG), similar to findings in gastric and duodenal specimens from patients with gastritis and ulcer[51]. The increase and remodeling of phospholipids with more percent arachidonic acid might be less protective of the gastric mucosa in addition to potentially yielding more arachidonic acid derived proinflammatory eicosanoids[67]. In contrast, higher PG, an inflammation-responsive lipid, reduces activation of toll-like receptors[68].

The association of low CD36 mRNA with gastric ulcer, gastritis, duodenitis, and gastrointestinal hemorrhage in the BioVu database suggests CD36 is critical for human gastric homeostasis. The PrediXcan approach, which examines the genetic component of gene expression using models trained with reference transcriptome data, increases the likelihood that associations observed between the expression of a gene and phenotypes are causal[52,53]. This interpretation is further buttressed by the identification of a previously unstudied CD36 SNP that strongly associates with death from gastrointestinal hemorrhage in the UK Biobank, providing additional support for relevance of the CD36 gene to gastrointestinal pathology. The SNP would disrupt the site directing a methylated histone H3K4me1 that regulates cell-identity-specific CD36 expression[69]. This H3K4me1 site is close to the most abundant CD36 promoter. Most of the enzymes that modify chromatin depend on intermediary metabolites functioning as substrates or cofactors, which helps explain why metabolism is critical for cellular renewal and differentiation after injury[70]. Thus, reduction in CD36-mediated FA metabolism would impair the chromatin modification needed for PC progenitor differentiation to PC and mucosal renewal (Fig. 9).

Data in rodents link CD36 to intestinal function and barrier integrity[24,25] and this study presents evidence of its important role in stomach tissue repair. Appropriate repair is required to

prevent gastritis, ulcers, hemorrhage, and the development of metaplasia[14]. The CD36 gene is polymorphic and CD36 variants associate with abnormal lipid metabolism[57] and risk of cardio-metabolic diseases[71,72]. Our findings in mice and biomedical databases support the contribution of CD36 variants to human gastrointestinal disorders. An estimated 11% of the US population suffer from a chronic digestive ailment, with prevalence reaching 35% for those 65 years and older and resulting in substantial morbidity, mortality, and cost[73].

## Methods

**Animals**. All protocols were approved by Washington University animal ethics committee. Mice housed in a 12 h light-dark facility were fed chow ad libitum (Purina) or 12 h fasted. Parietal cell Cd36 deficiency (PC-$Cd36^{-/-}$) was obtained by crossing *Cd36* floxed (Fl/Fl) mice with mice expressing the $H^+$-$K^+$-ATPase β ($Atp4b$) Cre. *Cd36* deletion in endothelial cells (EC-$Cd36^{-/-}$) used Tie2 Cre mice[24,46]. Cre$^+$ males were bred to Cre$^-$ females to avoid germline transmission. Genotypes were confirmed by PCR and immunohistochemistry. All studies used C57Bl6, sex-matched mice cohorts.

**Plasma hormones**. Plasma gastrin in 12 h fasted and 4 h refed mice was assayed by ELISA (RayBio). Leptin was measured in 12 h fasted and 4 h refed mice, and in the vagus-dependent pre-absorptive phase from plasma collected at 0 and 15 min after oral fat exposure using Milliplex mouse metabolic hormone magnetic bead panel (Millipore Sigma).

**High-resolution respirometry**. Isolated corpus mucosa from WT and $Cd36^{-/-}$ mice ($n = 4$/genotype) was permeabilized (50 μg/mL saponin, 30 min, 4 °C), washed (10 min, 4 °C) in Mir05 respiration buffer, blotted, microbalance weighed, and transferred in fresh Mir05 with 3 mg/mL creatinine to the Oxygraph-O2k (Oroboros) chamber. Non-phosphorylative "Leak" was measured in presence of malate (1 mM), glutamate (10 mM) and pyruvate (5 mM). Oxygen consumption in the oxidative phosphorylation state (OxPHOS) was assayed by adding ADP (5 mM) and succinate (10 mM) and maximum flux (ETS) by uncoupling with FCCP. Assays were in duplicate.

**Metabolomics and lipidomics**. Metabolites (including acetylcholine and acylcarnitines) and lipids were analyzed using liquid chromatography-tandem mass spectrometry as described[74] using the methanol/methyl *tert*-butyl ether/water extraction protocol[75]. Extracts were processed for lipidomic or metabolomic profiling.

**Fatty acid uptake**. Mice (12 h fasted, $n = 6$–7/genotype) were injected retro-orbitally with 100 μL PBS containing 2 μCi of [$^3$H]oleic acid (Perkin Elmer) and perfused with 10 mL PBS through the left ventricle 5 min after injection. Excised tissues were homogenized in 500 μL PBS and aliquots (100 μL) added to 5 mL scintillation fluid (Optiphase HiSafe 3, Perkin Elmer) for radioactivity measurement. Tissue FA uptake was adjusted by tissue weight and by antrum radioactivity which did not differ between groups.

**High-dose tamoxifen injury**. Tamoxifen treatment was as previously[42]. Tamoxifen (TAM) was injected intraperitoneally daily for 3 days (250 mg/kg). Stomachs were collected at day 1 or 5 postinjection and from untreated mice for immunofluorescence. For quantification of injury and recovery gastric sections from control and tamoxifen (TAM) treated mice, were processed for immunofluorescence and at least five images with 10 well-oriented gastric units were obtained per stomach. Parietal cells were immunostained for ezrin and VEGFB and proliferating cells for Ki67. Chief cells were co-stained with the neck cell marker Griffonia Simplicifolia Lectin II (GSII), referred to as spasmolytic polypeptide (TFF2) expressing metaplasia (SPEM) cells. Parietal cell number and diameter were quantified by ImageJ software and proliferating (Ki67$^+$), GIF$^+$/GSII$^+$ SPEM cells and PCs per gland unit were averaged for all units counted.

**Histology**. Stomachs were fixed in 10% formalin, and paraffin embedded[76]. Sections (5 μm) deparaffinized, treated (99 °C, 18 min) for antigen retrieval (rodent decloaker, Biocare Medical) were blocked 1 h in 1% donkey serum (Jackson Labs), 1% BSA (Sigma-Aldrich) and 0.3% Triton-X100. Primary antibodies (overnight, 4 °C) were followed by horseradish peroxidase (Jackson Immuno Research Labs) or fluorescently labeled secondary antibodies (Alexa Fluor 488 and 594; Invitrogen). Supplementary Table 1 lists primary antibodies and dilutions. Fluorescence used a Zeiss Axiovert 200 microscope with Axiocam MRM camera and Axiovision software. Bright-field images used a 2.0 HT NanoZoomer (Hamamatsu) whole slide scanner or an Olympus BX43 light microscope.

**Electron microscopy**. Stomachs, fixed overnight at 4 °C in modified Karnovsky's buffer, were sectioned into rings and processed for EM. Mitochondrial morphology was evaluated using ImageJ software (> 50 mitochondria/cell).

**Gene expression**. Total RNA, isolated with TRIzol (Invitrogen) from frozen stomachs ($n = 5$-6/genotype) was reverse transcribed (ABI Prim 7000 Sequence Detection System; Applied Biosystems). Quantitative real-time PCR used Power SYBR Green Master Mix and 7500 Fast Real-Time PCR System (Applied Biosystems). Relative mRNA content (housekeeping gene 36B4) used δCt calculations (primers in Supplementary Table 2).

**Western blotting**. Tissue proteins separated by SDS-PAGE (4–12% acrylamide; Invitrogen), transferred to polyvinylidene fluoride membranes (Millipore) and blocked (Li-COR Biosciences, Lincoln, NE), were incubated overnight (4 °C) with primary antibodies; mouse anti-mouse β-actin (1:5000), intrinsic factor (1:1000) (Santa Cruz Biotech), goat anti-mouse CD36 (1:1000, R&D Systems). Infrared dye-labeled secondary antibodies were added (1 h, room temperature) and signal detected (Li-Cor Odyssey Infrared) and adjusted to β-actin and total protein (Revert protein stain kit, Li-COR).

**Statistics and reproducibility**. Statistical analyses were performed using Graph-Pad Prism (V9; La Jolla CA). All data are means ± SEM. Two-way ANOVA followed by *post hoc* tests by Sidak multiple comparisons were used throughout, except as noted. Log-transformed metabolomic and lipidomic data were analyzed for effects of genotype, fasting, refeeding by two-way ANOVA using MetaboAnalyst 4.0 (FDR: 0.05) or in select cases using Prism 9. All statistical tests were two sided, with a $P$ value of 0.05 considered statistically significant and $P < 0.10$ indicating a trend.

**PrediXcan analysis**. PrediXcan[77] was used to evaluate CD36 contribution to the etiology of gastric disorders. The genetically determined expression component was estimated using gene expression imputation trained with reference transcriptome data (v6) from the Genotype-Tissue Expression (GTEx) Consortium[52,53]. PrediXcan was applied to patients of European ancestry with electronic health records tied to whole-genome genetic information in the BioVU resource of Vanderbilt University's[54]. Table 1 lists the number of cases and controls for each trait. PrediXcan analysis on BioVU data was deemed non-human subjects research by VUMC IRB# 151187.

**Reporting Summary**. Further information on research design is available in the Nature Research Reporting Summary linked to this article.

## Data availability
Source data files, including uncropped blots, are provided with this paper (Supplementary Data 1) and all other relevant data can be obtained from the corresponding author Miriam Jacome-Sosa upon reasonable request.

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

## Acknowledgements

We acknowledge assistance of the following facilities at Washington University: Nutrition and Obesity Research Center Cellular and Molecular Biology Core (NORC CMBC, P30 DK056341) with mitochondrial function and cytokine/hormone levels, the Hope Center Animal Surgery Core for pylorus ligation studies, the Hope Center Alafi Neuroimaging Laboratory (NIH Shared Instrumentation Grant S10-RR-027552) for Nanozoomer slide scanning and the Advanced Imaging and Tissue Analysis Core (AITAC, P30 DK052574) of the Digestive Disease Research Core Center (DDRC) for histology and microscopy. We acknowledge assistance of Karen Green at the Department of Pathology Electron Microscopy Center. We also acknowledge the Metabolomics Core Facility at the Physiology Institute of the Czech Academy of Sciences for lipidomics profiling. This work was supported by grants from the National Institute of Health DK060022 (NAA), DK060022S1 (MJS), DK094989, DK105129 (JCM), DK48370 and DK101332 (JRG), by NIH//NIA AG068026 and R01HG011138 (ERG) and a grant from Ministry of Education, Youth and Sports of the Czech Republic LTAUSA18104 (OK).

## Author contributions

M.J.S. designed experiments, obtained and analyzed data, and wrote the manuscript. Z.F.M., E.F.M., R.N., D.S., K.P., T.P., V.P., and H.G.L. helped with experiments and data analysis. AV conducted gastrin analysis, L.L.G. assisted with human data interpretation, J.R.G. contributed to data analysis and reviewed the manuscript, O.K. performed lipidomics and metabolomics and assisted with analyses, E.R.G. explored and analyzed human genetic data, J.C.M. contributed to experimental design, data analysis and reviewed the manuscript, N.A.A. designed experiments, analyzed data, and wrote the manuscript. M.J.S., J.C.M., and N.A.A. are guarantors of this work and, as such, had full access to all data and take responsibility for data integrity and accuracy of data analysis.

## Competing interests

The authors declare no competing interests.
