## [Transparent Peer Review File · Communications Biology]

Reviewers' comments:

Reviewer #1 (Remarks to the Author):

The manuscript shows the gastric morphology, matrix accumulation, inflammation, fatty acid uptake, and gastric secretions in CD36 double knockout mice. Furthermore, the author also found a high association of CD36 SNP (rs144921258) in patients with gastrointestinal hemorrhage. This study found relationships between CD36 and gastric functions. CD36 has been demonstrated in regulating fatty acid uptake, olfactory detection, and inflammatory response (J Clin. Invest. 2005; 115: 3177–3184 & Nat. Commun. 2020; 11:4765.). Here, a large number of results about fatty acid uptake were determined in CD36 deletion mice, but the rationale between fatty acid uptake and gastric function was unclearly described. In addition, the levels of gastrin and somatostatin were significantly changed in CD36^{-/-} mice compared with WT mice but gastric acid secretion was not altered in both mice. The authors also explored the proliferation of epithelial cells in both mice via the high-dose tamoxifen (TAM) injury model. However, the proliferation rate was similar in both CD36^{-/-} mice and WT mice.

Overall, the investigation was not dealt with underlying mechanisms of CD36 on gastric mucosa maintenance and in the absence of a solid conclusion.

Specific comments:

1. The CD36 is expressed in gastrin cells (Fig. 1E). Gastrin cells only exist in the antrum, however, the protein levels of CD36 were low in the antrum (Supplementary Fig. 1B). The data was a contradiction.
2. The colocalization between CD36 and GIF staining is found in the bottom left of Supplementary Fig. 1D, indicating chief cells also express CD36. That should be rechecked.
3. Leptin and ghrelin are two hormones that have opposite functions in response to food intake. Why the levels of leptin and ghrelin were all reduced in CD36 deletion (Fig. 1I-J)? In addition, the author's previous report showed that CD36-facilitated fatty acid uptake inhibits leptin production (Diabetes. 2007; 56: 1872-80.). Why the inconsistent results of leptin expression were observed in this study?
4. In Fig. 2, the authors implied that olfactory detection was regulated by CD36. However, a previous report has evidenced that CD36 is involved in olfactory detection of dietary lipids, spontaneous fat preference, and digestive secretions (J Clin Invest. 2005; 115: 3177–3184.)
5. CD36 binds many ligands including fatty acids and collagen. It's expected that stomach FA uptake was reduced by CD36 deletion (Fig. 5-7). However, why the collagen 1 α was deposited in CD36^{-/-} mice compared to WT mice (Fig. 3B)?
6. Activation of Smad1 is required for parietal cell differentiation. It's recommended determining parietal cell differentiation markers between CD36^{-/-} and WT groups directly in the evaluation of parietal cell progenitor ability.

Reviewer #2 (Remarks to the Author):

The gastric epithelium is often exposed to injurious elements and failure of appropriate healing predisposes to ulcers, hemorrhage, ultimately cancer. We examined gastric function of CD36, a protein linked to disease and homeostasis. Jacome-Sosa and colleagues have employed mouse genetics to analyze the functional relevance of CD36 in gastric epithelial homeostasis. They found that Stomachs of Cd36^{-/-} mice have altered gland organization and secretion, more fibronectin and inflammation. Further analyses show that a CD36 variant predicted to disrupt an enhancer site associates ($p < 10^{-17}$) to death from gastro-intestinal hemorrhage in the UK Biobank. These exciting

findings support role of CD36 in gastric tissue repair, its deletion associating with chronic diseases that can predispose to malignancy, which serves diversified interest of this journal's readership. This manuscript should be accepted for publication if following points are addressed.

1. One of the exciting discoveries of this study is the presence of CD36 in gastric epithelial cells. However, current study did not address the precise distribution pattern of CD36 in specific population of gastric glandular cells. Is CD36 enriched in gastric parietal cells? Where is CD36 specifically located within gastric parietal cells or chief cells?
2. For the general readership, what is the functional relevance of CD36 in gastric epithelial cell renewal/repair?
3. Fig.1C: The localization of ezrin in the top mold is not clear.
4. Fig 1K: Both of the plasma gastrin levels in fasted and refed condition need be reported.
5. Fig 4A-C: The gain value of representative histology and immunohistochemistry figure of CD36^{-/-} mice at day 5 post TAM seems to be different from other groups.
6. The statistical analysis diagram of the overall article needs more care such as standardization.

Reviewer #3 (Remarks to the Author):

The manuscript by Jacome-Sosa et al., entitled, "CD36 maintains the gastric mucosa and associates with gastric disease" examines the gastric function of CD36, a protein linked to gastrointestinal disease and tissue homeostasis. The authors found that CD36 knockout mice have altered gland organization and exhibit more fibronectin and inflammatory signaling. The authors also found that mucosal repair is abrogated in these mice, largely due to defects in progenitor cell differentiation that influence lipid metabolism and epithelial cell renewal. They circle back at the end of the manuscript to provide data correlating low CD36 expression with a variety of gastric disease outcomes. The manuscript is well written, and the experiments follow a logical progression, making it easy for the reader to follow. I have only a few comments to improve the quality of the manuscript.

Figure 1. It would be helpful to include magnification bars in the inset panels on c and E. Also the inset panels in 1C are not aligned. Could the authors please specify in the figure legend what statistical test was performed?

Figure 2. For panels D, E, F, and H could the authors please add labels to the X axes.

Figure 3. Could the authors please add magnification bars to the CD36^{-/-} panels in 3A? Could the authors please clarify why laminin is elevated in CD36 knockout mice under both fasting and refed conditions but fibronectin and collagen differ in this response? Have the authors confirmed any of the expression assays in Figure 3D at the protein level?

Figure 4. It might be helpful for the authors to quantify not just the incidence of proliferating cells, but also the amount of Ki67 staining using H-DAB plugin on ImageJ.

Figure 5G again the micrograph panels are not well-aligned. 5H the authors should label the X axes. Throughout the figures, the asterisks to signify statistical significance are hard to see, perhaps increase the size on those. Also please provide magnification bars on all micrographs, and align the panels.

Statistical analyses. The authors indicate that all data are compared via unpaired t test except when noted. It might be helpful to apply one-way ANOVA to these data as well. Similarly, log-transformations could be analyzed using non-parametric tests such as Kruskal-Wallis or Mann-Whitney U.

Can the authors clarify in the methods how many patients total were assessed? Is there overlap between the 184 patients who have gastric ulcer and 744 patients who have gastritis?

Have the authors tried any in vitro assays using snp rs144921258? This is outside of the scope of the current study, but would be highly informative to correlate the phenotypes seen in vitro with the disease outcomes associated with that snp.

Reviewers' comments

Referee #1

The manuscript shows the gastric morphology, matrix accumulation, inflammation, fatty acid uptake, and gastric secretions in CD36 double knockout mice. Furthermore, the author also found a high association of CD36 SNP (rs144921258) in patients with gastrointestinal hemorrhage. This study found relative ships between CD36 and gastric functions. CD36 has been demonstrated in regulating fatty acid uptake, orosensory detection, and inflammatory response (J Clin. Invest. 2005; 115: 3177–3184 & Nat. Commun. 2020; 11:4765.). Here, a large number of results about fatty acid uptake were determined in CD36 deletion mice, but the rationale between fatty acid uptake and gastric function was unclearly described. In addition, the levels of gastrin and somatostatin were significantly changed in CD36^{-/-} mice compared with WT mice but gastric acid secretion was not altered in both mice. The authors also explored the proliferation of epithelial cells in both mice via the high-dose tamoxifen (TAM) injury model. However, the proliferation rate was similar in both CD36^{-/-} mice and WT mice.

Overall, the investigation was not dealt with underlying mechanisms of CD36 on gastric mucosa maintenance and in the absence of a solid conclusion.

- We appreciate the Reviewer's general comments: We have revised the text to further clarify the rationale for linking fatty acid uptake and gastric function. We also addressed the comment about the mechanism for CD36 effect on gastric function by including a summary at the beginning of discussion that highlights the mechanistic findings of the study. All revisions to the text are highlighted in red.

-Rationale added to Introduction (Pages 3-4): Although cellular metabolism influences cell fate and function and is dysregulated in disease (References 15-17), its role in maintaining the gastric epithelium is unclear. Fatty acid uptake and FA oxidation have been shown important for tissue maintenance and repair most notably by keeping a competent niche of stem cells (PMID: 29727683, PMID: 31759926). Although FA oxidation does not acutely fuel proliferation of stem cells it is critical for maintaining the stem cells' long-term ability to differentiate (References 18 and 19). The scavenger receptor CD36 was reported abundant in stomach tissue (Reference 20) and its mRNA identified a decade ago in a microarray of magnetic bead sorted parietal cells (Reference 21). CD36 facilitates FA uptake and FA oxidation so we investigated if CD36-mediated FA metabolism is important for function and repair of the gastric mucosa.

- Mechanistic findings in this study (added to beginning of Discussion page 14): The major contributions of the work are the following: First, we show that Cd36 deletion impairs gastric function, metabolism and mucosal renewal after injury, through reduced endothelial cell delivery of fatty acids to the corpus and not by damaging parietal cell function. Second, as with other stem cell populations, the long-term defect in FA uptake and oxidation was found to result in less competent progenitor cells that are not amenable to full differentiation into parietal cells impeding recovery from injury. Defective release of leptin by the CD36 deficient stomach is another potential mechanism for the impaired gastric recovery that will need future study. Finally, the clinical impact of our work is highlighted by the finding that the defect of gastric recovery in low CD36 expression might be an underlying mechanism in the etiology of ulcer, gastritis and gastrointestinal hemorrhage.

Specific comments:

1. *The CD36 is expressed in gastrin cells (Fig. 1E). Gastrin cells only exist in the antrum, however, the protein levels of CD36 were low in the antrum (Supplementary Fig. 1B). The data was a contradiction.*

- The data are not contradictory for the following reasons: In the stomach CD36 is expressed in epithelial cells, endothelial cells and enteroendocrine cells. The difference between corpus and antrum can be related to the following: In the corpus CD36 is present in parietal cells, the predominant cell type (~70% of the mucosa) and is very abundant in endothelial cells and the corpus is highly vascularized. CD36 is also present in enteroendocrine cells which, as in the small intestine, are a small percent (1-2%) of total cells in the mucosa so enteroendocrine cell CD36 expression does not significantly alter total expression levels. The antrum lacks parietal cells and is less vascularized than the corpus. In the antrum CD36 is present on endothelial cells and on enteroendocrine cells. The enteroendocrine cells being a small percent of total cells, the major reason for the lower CD36 expression in the antrum is the lower vascularization in addition to the absence of parietal cells. This can be seen in Figure 1E where CD36 is present on endothelial cells (red) and gastrin expressing cells (green) are less than 1 per gland on average. So, the data are consistent with CD36 cellular distribution and not contradictory. We have included a statement related to this point in pages 5-6, lines 105 to 110.

2. The colocalization between CD36 and GIF staining is found in the bottom left of Supplementary Fig. 1D, indicating chief cells also express CD36. That should recheck.

- We thank the reviewer for the opportunity to elaborate on this issue. We realize we did not cite the extensive previous work in the lab of one of the authors (J Mills) showing that among epithelial cells, CD36 is exclusively expressed by parietal cells. J. Mills first identified CD36 as a parietal-cell specific gene using magnetic-bead-sorted parietal cells (PNAS, 2001 PMID: 11717430). His group followed up on this using laser-capture microdissected cells in (Physiological Genomics PMID: 19208773) and most recently, using mice with lineage-specific expression of GFP specifically in parietal cells to flow-sort parietal cells from other cell types (Genes & Development, PMID: 28174210). CD36 expression was nearly undetectable in non-parietal cells and the 3rd highest signal of all transcripts in parietal cells (100-fold enrichment). Similarly, localizing CD36 using immunohistochemical and immunofluorescent techniques for two decades always shows the same pattern of endothelial and epithelial CD36 signal specifically in parietal cells at their base, where their plasma membrane abuts the basement membrane. We do not see chief cell localization by any method and are unsure where the reviewer is seeing that pattern in our data. Possibly, the reviewer is referring to parietal cells in the base of gastric units that are mixed in amongst the chief cells, so sometimes the picture can be confusing.

In the revised manuscript we have added text and references detailing the various approaches (magnetic beads, laser-capture, lineage-based fluorescent sorting) previously used to confirm parietal cell expression of CD36 (page 4 lines 74-75 and page 5 lines 96 to 100). We also have provided a magnification of Supplementary Fig 1D to show that CD36 is on parietal cells and not on chief cells at the base of the glands (Supplementary Fig. 1D).

3. Leptin and ghrelin are two hormones that have opposite functions in response to food intake. Why the levels of leptin and ghrelin were all reduced in CD36 deletion (Fig. 1I-J)? In addition, the author's previous report showed that CD36-facilitated fatty acid uptake inhibits leptin production (Diabetes. 2007; 56: 1872-80.). Why the inconsistent results of leptin expression were observed in this study?

- While we show both leptin expression and plasma levels (Fig.1I and K), we only show ghrelin expression (Fig.1H), as its plasma levels were highly variable. Because of this, the different forms of circulating ghrelin and the difficulty of accurately measuring active ghrelin, its actual levels (which we could not measure) might not have followed expression levels, thus it cannot

be inferred that the two hormones varied in the same or opposite direction. A more important point that needs to be factored in is the different regulation of adipose versus stomach leptin as explained below:

- With respect to the role of CD36 in leptin secretion, as the reviewer noted, we previously showed that fatty acid uptake in CD36 deficiency increases leptin secretion from adipose tissue as also later reported by Cammisotto et al (PMID: 20621336). In contrast, in the present study, we find *Cd36* deletion decreases leptin response to refeeding and during the first 15 min after a fatty meal. As we noted in the manuscript and further clarify it in the revision, leptin secretion by adipose tissue is not regulated by the same mechanisms as gastric leptin. Adipose leptin does not respond rapidly to food intake, its release being slow requiring more than 60 min for cells to increase leptin synthesis and then secretion (PMID: 20621336). On the other hand, food intake through vagal input acutely releases gastric leptin by chief cells (see page 6, lines 118-121). So after feeding, the change in circulating leptin reflects the effect on gastric leptin, and depending on basal leptin levels the increase derived from the stomach might or might not be detected. The marked reduction in leptin secretion consequent to low vagal input would impair epithelial healing (References 61-62) and could have contributed to the defect in mucosal recovery from injury. A paragraph addressing this point has been added to pages 15 line 309 to 319)

- Although not relevant to the present study, we addressed the question of the reviewer related to the effect of fatty acids on leptin. We conducted an experiment to directly compare the response of leptin secretion from stomach and adipose tissue explants to added fatty acids using the same experimental conditions and the same mice. We found that leptin secretion from *Cd36*^{-/-} stomachs is reduced by FAs but this is not the case for leptin from adipose tissue. The results from this experiment were not included since they do not pertain to the findings in this manuscript. They indicate that more work is needed to better understand the regulation of the two leptin sites and their metabolic implication.

Leptin secretion from stomach and adipose tissue explants.

Stomach and epididymal fat from three WT and four *Cd36*^{-/-} mice were minced. Explants were incubated in 2 ml of M199 with 10% fetal bovine serum under 95/5 O₂/CO₂ at 37°C for 20 h. The next day, medium was replaced and 100 μM oleic acid complexed with BSA (0.5:1) was added. After 6 h, the medium and explants were harvested and frozen at -80 °C until analysis. Leptin in the media was measured using a Milliplex mouse adipokine magnetic bead panel (Millipore Sigma) and normalized by tissue weight.

4. In Fig. 2, the authors implied that orosensory detection was regulated by CD36. However, a previous report has evidenced that CD36 is involved in orosensory detection of dietary lipids, spontaneous fat preference, and digestive secretions (*J Clin Invest.* 2005; 115: 3177–3184.)

- The reviewer is correct. The orosensory role of CD36 in perception of dietary lipids is well established as described in the review we cited in the manuscript (Besnard P *Rev Endocr Metab Disord* 2016). We have expanded on this in the text and included the original reference in addition to the review (page 6 lines 122 to 125). The finding that *Cd36* deficiency influences the cephalic leptin response of the stomach has not been reported and is relevant since defective stomach leptin secretion could contribute to the impaired mucosal recovery from injury.

5. *CD36 binds many ligands including fatty acids and collagen. It's expectable that stomach FA*

uptake was reduced by CD36 deletion (Fig. 5-7). However, why the collagen 1 α was deposition in CD36^{-/-} mice compared to WT mice (Fig. 3B)?

- We find increased fibronectin but not increased collagen deposition in Cd36^{-/-} stomachs. This is consistent with our previous findings in the small intestine. In the initial version we showed a small decrease in collagen deposition in refed mice. Although fasting and refeeding might differentially alter components of the extracellular matrix, for simplicity we added new data collected in nonfasted mice, which would be a steady state condition as opposed to prolonged fasting and acute refeeding. This showed that the increase in matrix is specific to fibronectin (pages 7- 8, lines 150-154, and Fig. 3B).

6. Activation of Smad1 is required for parietal cell differentiation. It's recommended determining parietal cell differentiation markers between CD36^{-/-} and WT groups directly in the evaluation of parietal cell progenitors ability.

- We agree with the reviewer that signaling through SMADs, in particular via BMPR1A and the BMP2/4/7 growth factors, influences parietal cell differentiation; however, parietal cell progenitors are rare and have no specific markers (see recent publication by one of the authors, J Mills on how parietal cell differentiation is regulated: PMID 33280496). The nature and molecular signaling/markers that govern parietal cell emergence from stem cells is of intense interest to the Mills Lab, and it is a complex question that we hope to address in upcoming manuscripts. Unfortunately, although BMP signaling via SMADs is required for normal parietal cell maintenance, this has not translated into an easy way to mark parietal cell progenitors for several reasons in addition to their scarcity. For one, it is not the mere expression of SMAD1 which is important, but its localization and phosphorylation are critical. In addition, SMAD1 itself is not expressed only in parietal cells but in chief cells as well (e.g., see Protein Atlas: <https://www.proteinatlas.org/ENSG00000170365-SMAD1/tissue/stomach#img>). In short, we are working on this difficult and fascinating issue, but there is no way to currently address the concept of differences in parietal cell progenitors with respect to CD36 requirement and expression.

Referee #2

The gastric epithelium is often exposed to injurious elements and failure of appropriate healing predisposes to ulcers, hemorrhage, ultimately cancer. We examined gastric function of CD36, a protein linked to disease and homeostasis. Jacome-Sosa and colleagues have employed mouse genetics to analyze the functional relevance of CD36 in gastric epithelial homeostasis. They found that Stomachs of Cd36^{-/-} mice have altered gland organization and secretion, more fibronectin and inflammation. Further analyses show that a CD36 variant predicted to disrupt an enhancer site associates ($p < 10^{-17}$) to death from gastro-intestinal hemorrhage in the UK Biobank. These exciting findings support role of CD36 in gastric tissue repair, its deletion associating with chronic diseases that can predispose to malignancy, which serves diversified interest of this journal's readership. This manuscript should be accepted for publication if following points are addressed.

- We thank the Reviewer for the positive comments on the exciting findings of our study. We have addressed all the concerns of the reviewer as detailed below. All revisions to the text are highlighted in red.

1. One of the exciting discoveries of this study is the presence of CD36 in gastric epithelial cells. However, current study did not address the precise distribution pattern of CD36 in specific

population of gastric glandular cells. Is CD36 enriched in gastric parietal cells? Where is CD36 specifically located within gastric parietal cells or chief cells?

We thank the reviewer for the opportunity to elaborate on this issue. We realize we did not cite the extensive previous work in the lab of one of the authors (J Mills) showing that among epithelial cells, CD36 is exclusively expressed by parietal cells. J. Mills first identified CD36 as a parietal-cell specific gene (PNAS in 2001 (PMID: 11717430) using magnetic-bead-sorted parietal cells. His group followed up on this using laser-capture microdissected cells (Physiological Genomics, PMID: 19208773) and most recently, using mice with lineage-specific expression of GFP in parietal cells to flow-sort parietal cells from other cell types (Genes & Development, PMID: 28174210). CD36 expression was nearly undetectable in non-parietal cells and the 3rd highest signal of all transcripts in parietal cells (100-fold enrichment). Similarly localizing CD36 using immunohistochemical and immunofluorescent techniques for two decades always shows the same pattern of endothelial and epithelial CD36 signal specifically in parietal cells at their base, where their plasma membrane abuts the basement membrane.

In the revised manuscript we have added text and references detailing the various approaches (magnetic beads, laser-capture, lineage-based fluorescent sorting) previously used to confirm parietal cell expression of CD36 (page 4 line 75 and page 5 line 96 to 100). We also provide a magnification of Supplementary Fig 1D to show that CD36 is only on parietal cells and not on chief cells at the base of the glands (Supplementary Fig. 1D).

2. For the general readership, what is the functional relevance of CD36 in gastric epithelial cell renewal/repair?

In addition to summarizing our findings at the beginning of discussion, we have revised summary figure 9 to add the condition of injury and how CD36 deficiency impacts the process of recovery and increases risk of specific gastric diseases. We thought this would illustrate CD36 relevance to the general readership

3. Fig.1C: The localization of ezrin in the top mold is not clear.

- Thank you for pointing this out. We hope that the new revised image where the lumen is clearly visible, highlights better ezrin localization now in green at the apical side of parietal cells while CD36 expression is at the basolateral membrane and is excluded from the apical side.

4. Fig 1K: Both of the plasma gastrin levels in fasted and refed condition need be reported.

-We now include plasma gastrin levels in fasted and refed conditions, in Fig. 1J of the revised manuscript. Findings confirm the higher levels of plasma gastrin in non-fasted *Cd36^{-/-}* mice (data now excluded to avoid redundancy), particularly during refed conditions. We also revised the methods section accordingly (page 18, line 376-377).

5. Fig 4A-C: The gain value of representative histology and immunohistochemistry figure of CD36^{-/-} mice at day 5 post TAM seems to be different from other groups.

- We revised this figure, and we confirm that the gain value of representative images is the same. However, we picked another image, also with similar gain that could be more representative. We did notice this issue in appearance which may be due to the longer glands in recovered *Cd36^{-/-}* mice.

6. The statistical analysis diagram of the overall article needs more care such as standardization.

- We addressed this concern and similar concerns from Referee #3 regarding the statistical analysis. We re-analyzed all the data using the most appropriate statistical design for each set and we describe in every figure legend the test that was used. The two-way ANOVA analysis was applied to most of the data and we focused on the relevant outcomes, as detailed in the list below. We also revised the statistical analysis section to give a more detailed description (page 21, lines 434-440). The two-way ANOVA adjustment resulted in some differences in *P* values that did not change data interpretation.

- A. We removed gastrin releasing peptide (GRP) mRNA levels from Fig.1 and line 113, as they missed significance by two-way ANOVA.
- B. Figure 2 A-C. We focused our two-way ANOVA analysis of the data on plasma hormones to two time points; 0 and 15 min after the fatty meal.
- C. In addressing a comment from Referee#3, we conducted new measurements of extracellular matrix markers using stomachs from nonfasted mice at steady state. As before, we found that fibronectin increased but neither laminin nor collagen levels differ between WT and *Cd36*^{-/-} stomachs. Thus, Fig. 3 and pages 7-8, lines 150-153 were revised accordingly.
- D. In Fig. 3, some changes in *P* values for inflammatory markers were observed after two-way ANOVA analysis and figures and text were updated accordingly. Overall only one marker (MCP1) had to be excluded for missing significance. Data interpretation was not revised.
- E. Fig. 4, in the tamoxifen studies, we removed the TAM 3 day group from the analysis and included it in the Supplement as a separate figure (now Supplementary Fig. 2). This was based on the well reported effect of TAM on PC death by numerous studies (references 41 and 42) and we did not observe any differences between genotypes. The TAM recovery data relative to control groups was then analyzed by two-way ANOVA, as suggested.
- F. Expression of DGAT1 was not different between groups by two-way ANOVA while that of AGPAT1 in the same pathway was, as it increases with refeeding in WT but not *Cd36*^{-/-} stomachs. Fig. 6 panel E now includes AGPAT and the text was revised accordingly (page 10, lines 217-218).
- G. We removed mRNA levels of Tff2 from Supplementary Fig.1G, as its levels were not changed after refeeding by two-way ANOVA. Sentences describing Fig. 1G were revised due to changes in *P* values after two-way ANOVA analysis in all cell markers (page 6, lines 112-113).
- H. Overall, all figures were revised to include changes in *P* values. We also added information on the statistical test that was performed to all figure legends.

Referee #3

The manuscript by Jacome-Sosa et al., entitled, "CD36 maintains the gastric mucosa and associates with gastric disease" examines the gastric function of CD36, a protein linked to gastrointestinal disease and tissue homeostasis. The authors found that CD36 knockout mice have altered gland organization and exhibit more fibronectin and inflammatory signaling. The authors also found that mucosal repair is abrogated in these mice, largely due to defects in progenitor cell differentiation that influence lipid metabolism and epithelial cell renewal. They circle back at the end of the manuscript to provide data correlating low CD36 expression with a

variety of gastric disease outcomes. The manuscript is well written, and the experiments follow a logical progression, making it easy for the reader to follow. I have only a few comments to improve the quality of the manuscript.

We thank the Referee for the positive comments regarding the study and clarity of the manuscript. We have endeavored to address all the points raised as described below. All revisions to the text are highlighted in red.

1. Figure 1. It would be helpful to include magnification bars in the inset panels on C and E. Also the inset panels in 1C are not aligned. Could the authors please specify in the figure legend what statistical test was performed?

- Magnification bars have been added to all images and inserts. We also added information on the statistical test that was performed to all figure legends.

2. Figure 2. For panels D, E, F, and H could the authors please add labels to the X axes

- Thank you for pointing this out. Labels have been added to all X axes.

3. Figure 3. Could the authors please add magnification bars to the CD36^{-/-} panels in 3A? Could the authors please clarify why laminin is elevated in CD36 knockout mice under both fasting and refeed conditions but fibronectin and collagen differ in this response? Have the authors confirmed any of the expression assays in Figure 3D at the protein level?

- Magnification bars have been added to all images.
 - With respect to laminin and other ECM proteins, we agree with the reviewer that changes in levels of these proteins between fasting and refeeding conditions are unnecessarily confusing. To address, we conducted new measurements using stomachs from nonfasted mice, which would steady state as compared to prolonged fasting and acute refeeding. We found that neither laminin nor collagen levels differ between WT and *Cd36^{-/-}* stomachs. Fibronectin remained consistently elevated in knockout mice (page 7-8, lines 150-154, Fig. 3B).

4. Figure 4. It might be helpful for the authors to quantify not just the incidence of proliferating cells, but also the amount of Ki67 staining using H-DAB plugin on ImageJ.

- We appreciate the suggestion of the referee but since Ki67 expression is a marker of mitotic cells, we used it as a yes-no metric to determine if cells are mitotic or not. We consider that assessing the number of cells that are in the cell cycle would be more reliable than assessing changes in intensity of Ki67 which differs in G1, S, G2 or M phases. We are unsure as to what additional relevant information would be garnered by knowing if cells are expressing a larger relative amount of Ki67.

5. Figure 5G again the micrograph panels are not well-aligned. 5H the authors should label the X axes.

- See below.

6. Throughout the figures, the asterisks to signify statistical significance are hard to see, perhaps increase the size on those. Also please provide magnification bars on all micrographs, and align the panels.

- All figures have been revised to include magnification bars and to increase the font size.

7. Statistical analyses. The authors indicate that all data are compared via unpaired t test except when noted. It might be helpful to apply one-way ANOVA to these data as well. Similarly, log-transformations could be analyzed using non-parametric tests such as Kruskal-Wallis or Mann-Whitney U.

We re-analyzed all the data using the most appropriate statistical design for each set and we describe in every figure legend the test that was used. We confirmed that all variables were normally distributed, for which we used two-sided, unpaired t –tests. Log transformations were performed on metabolomics and lipidomic data within the MetaboAnalyst workflow routine. The two-way ANOVA analysis was applied to most of the data and we focused on the relevant outcomes, as detailed in the list below. We also revised the statistical analysis section to give a more detailed description (page 21, lines 434-440). The two-way ANOVA adjustment resulted in some differences in *P* values that did not change data interpretation.

- A. We removed gastrin releasing peptide (GRP) mRNA levels from Fig.1 and line 113, as they missed significance by two-way ANOVA.
- B. Figure 2 A-C. We focused our two-way ANOVA analysis of the data on plasma hormones to two time points; 0 and 15 min after the fatty meal.
- C. In addressing a comment from Referee#3, we conducted new measurements of extracellular matrix markers using stomachs from nonfasted mice at steady state. As before, we found that fibronectin increased but neither laminin nor collagen levels differ between WT and *Cd36*^{-/-} stomachs. Thus, Fig. 3 and pages 7-8, lines 150-153 were revised accordingly.
- D. In Fig. 3, some changes in *P* values for inflammatory markers were observed after two-way ANOVA analysis and figures and text were updated accordingly. Overall only one marker (MCP1) had to be excluded for missing significance. Data interpretation was not revised.
- E. Fig. 4, in the tamoxifen studies, we removed the TAM 3 day group from the analysis and included it in the Supplement as a separate figure (now Supplementary Fig. 2). This was based on the well reported effect of TAM on PC death by numerous studies (references 41 and 42) and we did not observe any differences between genotypes. The TAM recovery data relative to control groups was then analyzed by two-way ANOVA, as suggested.
- F. Expression of DGAT1 was not different between groups by two-way ANOVA while that of AGPAT1 in the same pathway was, as it increases with refeeding in WT but not *Cd36*^{-/-} stomachs. Fig. 6 panel E now includes AGPAT and the text was revised accordingly (page 10, lines 217-218).
- G. We removed mRNA levels of Tff2 from Supplementary Fig.1G, as its levels were not changed after refeeding by two-way ANOVA. Sentences describing Fig. 1G were revised due to changes in *P* values after two-way ANOVA analysis in all cell markers (page 6, lines 112-113).
- H. Overall, all figures were revised to include changes in *P* values. We also added information on the statistical test that was performed to all figure legends.

8. Can the authors clarify in the methods how many patients total were assessed? Is there overlap between the 184 patients who have gastric ulcer and 744 patients who have gastritis?

- The total number of patients for each trait is the sum of cases and controls for each. There is overlap of 10 shared cases between gastric ulcer and gastritis duodenitis patients. This information is now included in the legend of Table 1.

9. Have the authors tried any in vitro assays using snp rs144921258? This is outside of the scope of the current study, but would be highly informative to correlate the phenotypes seen in vitro with the disease outcomes associated with that snp.

- We agree with the reviewer that this would be informative and hope to address it in future studies.

REVIEWERS' COMMENTS:

Reviewer #2 (Remarks to the Author):

The authors have taken my critiques with care and carried out requested experimentation. I am satisfied with the revision and therefore vote for acceptance.

Reviewer #3 (Remarks to the Author):

The authors thoughtfully and completely addressed all of my concerns and I endorse publication.